# Joint- or Crack-Opening Resistance Evaluation of Waterproofing Material and System for Structural Sustainability in Railroad Bridge Deck

**DOI:** 10.3390/ma13194229

**Published:** 2020-09-23

**Authors:** Kyu-Hwan Oh, Soo-Yeon Kim, Yong-Gul Park

**Affiliations:** 1Department of Railway Construction, Graduate School of Railway, Seoul National University of Science & Technology, Gongneung-ro, Nowon-gu, Seoul 01811, Korea; kyuhwan.oh@seoultech.ac.kr; 2Construction Technology Institute, Seoul National University of Science & Technology, Gongneung-ro, Nowon-gu, Seoul 01811, Korea; ksr1115@seoultech.ac.kr

**Keywords:** joint or crack opening, high-speed railroad bridge, waterproofing material, new evaluation method, bridge deck

## Abstract

A joint- or crack-opening resistance evaluation method for the selection of optimal waterproofing material for a railroad bridge deck is proposed. The joint opening range (mm) for the evaluation, and a typical load case of a high-speed double-track railroad bridge structure deck, is analyzed through the finite element method (FEM) and the results of analysis are used to calculate the minimum opening range. The evaluation method is then demonstrated with 4 commonly used waterproofing types of cementitious membrane system: a polyurethane coating system, self-adhesive asphalt sheet system and synthetic polymerized rubber gel composite asphalt sheet system. Five specimens of each type are subjected to continuous joint opening under 4 different joint width range conditions (1.5, 3.0, 4.5, and 6.0 mm), and the joint-opening resistance performance is compared. The proposal for the evaluation criteria and the specimen test results demonstrate how the evaluation method is pertinent for future selection of waterproofing membranes for the sustainability of high-speed railroad bridge deck structures.

## 1. Introduction

### 1.1. Background

As the expectation in the increase of vehicle dynamic stability performances has increased owing to the requirement for faster and heavier trains, new challenges to maintain the integrity of railroad structures constantly arise [1]. The results of resolving these challenges in the past have resulted in the state-of-the art high-speed railway lines that we are familiar with today, including Train de Grand Vitesse (TGV) in France, Shinkansen in Japan, Korea Train Express (KTX) in Korea, and China Railway High-speed (CRH) with the highest forms of railway technological innovations. High-speed railway lines are continuously in the process of expanding, and new railroad tracks and bridges are being constructed with higher standards of safety and stability [2]. The process of constructing large-scale railroad infrastructures and bridges over cities and populated areas often use pre-stressed concrete (PSC) panels. Steel-reinforced concrete railroad bridge decks, in particular, are exceptionally sensitive to penetration of water mixed with chloride-based compounds, and subsequently occurring steel corrosion will lead to cracks in the concrete substrate. In these structures, many expansion/construction joints and cracks are factors of structural degradation, reducing the service life of the railroad bridge [3]. Nielson et al. provide extensive research on the life-cycle cost estimation for railway bridge maintenance, and concrete deck cracking and spalling is a significant factor that affects the cost estimation for maintenance and repair of railway bridges [4]. Recently, there has been a noticeable focus on the installation of proper waterproofing systems to ensure the long-term sustainability of bridge structures against corrosion-induced cracking [5,6,7].

For bridge deck structures, manuals on waterproofing in concrete bridge decks exist in the US, Scotland, Canada, and UK (as well as others in different languages), and for railroad waterproofing, Dammyr et al. provides a comparison of Norwegian waterproofing and other general European waterproofing concepts in the context of securing higher performance waterproofing in railroad tunnels [8]. Numerous other research works indicate that waterproofing of highway bridges and railroad bridges is a topic of high interest in China, where high-performance waterproofing membrane types are regularly tested and reported in research papers. Song Liu et al. also provides research on the effect of mastic asphalt waterproofing on a high-speed railroad to prevent freeze-thaw damage [9]. Lai et al. provides a basic outline on performance requirement and application guideline of polyurethane waterproofing coating used for waterproofing intercity railway bridge decks and provides a case study of the Fo-Zhao intercity railway bridge to conclude that defects have not occurred since completion of construction [10]. Zhang provides an outline of the common problems found in bridge decks caused by leakage and lack of proper waterproofing and proposes selection criteria for a waterproofing layer method to prevent these common problems [11]. Xu proposes a spray-type polyurethane coating material quality specification and installation method for securing corrosion resistance of railway steel deck bridges [12].

Aside from durable concrete designs, waterproofing is the only other frontline for the protection of bridge decks against crack and water-leakage induced degradation of the structure, and conventional waterproofing systems are already being tested to ensure the installed waterproofing membrane will not degrade under exposure to temperature change or chemical substances [13]. However, a problem lies in that under the current specifications, the parameters for evaluating waterproofing materials concern the physical properties of the waterproofing material itself, and it is still not yet clear how to evaluate the “waterproofing performance” of the materials for selection. Although many waterproofing material products and types pass the minimum required performance criteria, some material types are not suitable for use in certain types of structure, one of these being railway bridges. Recent articles in the Rail Engineer and Paint Coating Industry discusses the weaknesses of few types of waterproofing material, particularly that of mopped emulsion-type waterproofing as it de-bonds from the bridge deck and allows water to diffuse underneath the waterproof layer and on the concrete deck [14]. The risks of de-bonding, membrane adhesion and cohesion failure, and crack bridging ability (crack-opening resistance) must be examined clearly before deciding to the use the waterproofing material, but the American Railway Engineering and Maintenance-of-Way Association (AREMA) or other railway codes do not specify this property as a requirement for waterproofing. As a matter of fact, waterproofing of railway bridges is often glossed over in these standard documents, only highlighting that waterproofing should be installed “carefully,” or “must be high performance,” without a specific guideline on how to evaluate the waterproofing material [15].

To properly assess and select a waterproofing system suitable for a railway bridge deck, a new evaluation method that tests the waterproofing material’s performance during service is required. In that regard, in this study, a joint opening resistance evaluation method of waterproofing systems that evaluates both the waterproofing property and sustainability against periodic opening is proposed. Waterproofing failure can be due to low general quality performance or improper workmanship during installation, and one aspect that is rarely discussed is that of maintenance of water tightness of the waterproofing system during joint or crack opening [16]. In the cases where a crack has formed on a concrete bridge deck or in a joint, stress will be localized at the joint/crack tip due to load during train operation, causing an existing joint/crack to open. Waterproofing membranes must be able to maintain adhesion on the concrete surface and prevent the joint/crack section from becoming a leakage path into the reinforced concrete matrix, otherwise cracks will eventually form on the underside of the bridge structure, accelerating the deterioration process and risking sustainability reduction. Refer to Figure 1 for case study of a bridge structure degradation status.

The subsequent sections will discuss the following: (1) in the discussions section, a theoretical crack or joint (hereby simplified to joint) opening mechanism and the necessity to develop a method for evaluating the joint-opening resistance performance of waterproofing membranes is outlined. (2) Next, a technical process for deriving the joint opening range parameters prior to evaluation is proposed. In this study’s case, a high-speed double-track railroad bridge structure designed in accordance with the Korean Railway Code (KR-C) 14030 railroad bridge deck specification is analyzed through the finite element method (FEM) and the deformation conditions are proposed as the requirement for setting the minimum opening range to be used during testing. (3) Based on the FEM analysis, the maximum deformation case is used as the minimum opening (higher opening ranges are included for demonstration purposes), and testing is conducted to demonstrate that waterproofing membranes can be comparatively evaluated based on the total number of opening cycles resisted until their maximum resistance capacity is reached (extrapolated by the point at which leakage occurs).

### 1.2. Waterproofing Systems to Protect the Degradation of Railroad Bridge Deck

It is common knowledge that concrete cracking is inevitable and the joints/cracks are pathways for chloride ions mixed with water to penetrate a concrete matrix and cause carbonation of concrete. Carbon dioxide (CO_2_) penetrates into concrete and reacts with the calcium to form calcite compounds (CaCO_3_), lowering the pH of concrete [17]. When the pH of concrete is lowered to approximately 9 the carbon steel rebar loses the passivating film and corrosion occurs, followed by rust formation. Since rust of the layer occupies a larger volume, it exerts internal pressure in the concrete matrix, causing spalling and further cracks. The main cause of corrosion-induced damage in bridge structures can be identified as due to failure of waterproofing performance as the primary function of the waterproofing layer is to prevent the contact of water with concrete. While minute cracking may be inevitable, prevention of further cracking in most cases can be resolved with a selection of optimal waterproofing material types. It is a common opinion that waterproofing performance failure is mainly attributed to poor workmanship during installation, but it must also be noted that a significant part of this issue is due to an improper understanding of the waterproofing membrane joint/crack resistance performance.

Most of the recently constructed railroad bridge decks consist of concrete plate structure comprised of a combination of concrete and steel rebars. Waterproofing membranes are most commonly installed between the rail overlay (the track bed) and the concrete or steel deck substrate. An illustration of waterproofing installation in a PSC bridge girder-based railroad bridge structure is shown in Figure 2.

While the slab thickness of the bridge deck is different depending on the bridge type, the PSC girder box construction specification in Korea states a height of 3.5~4 m, and the composition of the track bed about 1 m in thickness comprised of ballast layer (thickness: 300~350 mm) beneath the sleepers, ballast mat (thickness: 100~150 mm), protection concrete (thickness: 300~400 mm) and waterproofing layer (thickness: 2~4 mm). At the cross-sectional structure of the high-speed rail bridge, the bridge deck can be divided into two parts: track and sidewalk. Depending on the design standard, a water flow drainage system is installed at the respective sides of the track bed.

A specified list of waterproofing material types does not exist, but based on existing studies and reference materials (recommendations of successful types of waterproofing products exist in forms of articles and product catalogues), a generalized classification can be drafted. [18] Representative waterproofing systems for bridges can comprise: (1) sheet membrane system, (2) spray or liquid applied membrane system, and (3) cementitious slurry membrane system. A series of simplified illustrations of the waterproofing system schematic on railroad bridge decks is outlined in Table 1 and Table 2 shows the waterproofing systems used by respective nations/regions in the world.

Traditionally, waterproofing material type was selected based on the lowest price, easy workability, and/or short construction time but the requirement for selecting the materials based on their performance has taken priority over price [19]. However, due to the complexity of the degradation mechanism in the infrastructures, there is still a distinct lack of clarity as to which product and type to select based on the given environmental conditions [20]. Zhou and Xu provide a review of the various factors that affect the interface adhesion of waterproofing membranes on a concrete surface, including surface roughness, material property and thickness, and temperature [21]. Adhesion strength initially increases and decreases as the thickness of the membrane increases, and the same mechanism applies to surface roughness as well. High temperature during installation is also pertinent for securing long-term high adhesion performance as well. Ferdou et al. investigate an optimal mixture design for epoxy-based polymer for application in concrete structures [22], and Khotbehsara et al. provide an extensive study on the effect of high temperature on the mechanical properties of particulate-filled epoxy polymers used in infrastructures [23]. Su and Bloodworth also propose a numerical analysis method of sprayed-type waterproofing in tunnels that can simulate composite action, and while they provide a recommended design for spray-applied waterproofing, these studies attest that (1) waterproofing membranes must be properly evaluated against a myriad of factors, and most importantly against composite actions such as tension, compression and shear forces, and (2) a laboratory test method that can be used to evaluate these properties of waterproofing membranes is still in the process of being developed [24]. The literature review indicates that while there are numerous approaches towards understanding the degradation conditions, there is still a need to develop or improve the evaluation methods of waterproofing membranes. In light of this review of the literary background, this study proposes that a new joint-opening resistance evaluation method for waterproofing membranes can contribute towards improving the sustainability of railway bridge structures.

### 1.3. Existing Standardized Test Methods for Waterproofing Materials

Existing standardized test methods are not sufficient to completely simulate environmental degradation and concrete joint opening at the same time, and the criteria of evaluation only observes changes to physical properties rather what can be considered as the crucial ‘waterproofing’ property [25,26]. The waterproofing property (performance) is defined by the water tightness of the system against hydrostatic pressure, and the waterproofing system must be able to maintain this property against period degradation [27,28]. Visual observation or quantified assessment of specific performance criteria should be followed up with hydrostatic pressure resistance testing to see if a leakage path has formed within the installed system, but the existing evaluation methods are lacking in this method. International test methods today evaluate the durability of waterproofing membranes based on the following key parameters: tensile/tearing strength, adhesion strength, water pressure resistance, thermal resistance, chemical resistance (alkali and chlorine) and elongation [29]. The respective test methods and specifications outlined in Table 3 standards do not provide clear instructions on how to ensure proper workmanship, nor do they reflect on the waterproofing property of an installed system. In general, the standards recommend following the manufacturer’s specifications, but do not provide quality control instructions for each specification. Furthermore, and most importantly, aside from Japanese and US standards, a clear evaluation method on crack bridging/joint opening of waterproofing membranes is lacking [30], and even the evaluation methods employed in these standards are only based on visual observation for the final assessment. Table 3 lists specific reference materials from each national standard bodies.

## 2. Theoretical Discussion

### 2.1. De-Bonding and Cohesive Failure due to Zero-Span Tensile Stress

Cracks in a concrete member are subject to localized tension by stress concentration factor. When cracks form, the stress distribution is interrupted, and the stress at the anomaly section (the crack or discontinuity) is more concentrated than the surrounding region [31]. The stress concentration factor is commonly expressed by the following equation:(1)$$K_{t}=\frac{\sigma_{max}}{\sigma_{nom}}$$
where$K_{t}$ = stress concentration factor$\sigma_{max}$ = highest stress$\sigma_{nom}$ = nominal stress


While a break or fracture on most conventional waterproofing membrane types is uncommon, cyclic opening of the crack or joint section caused by bending moment following the train operation and steel reinforcement corrosion can cause the membrane to be subject to concentrated tension at areas with concrete crack opening [32]. The design guide by the Steel Construction Institute in the UK indicates that there is a strong probability of waterproofing failure occurring across the movement joint at the bridge deck sections [33]. Studies in Japan show that the localization of waterproofing membrane stress is at the tip of the concrete joint or crack (the section where opening occurs) [34]. Figure 3 illustrates the effect of zero-span tensile stress on the waterproofing membrane installed over concrete joints/cracks. When subjected to this cyclic stress, the membrane layer can potentially undergo various forms of fracture as the tensile stress acts on the adhesive bond between the membrane and the concrete surface, as well as and the cohesive bond within the membrane itself.

### 2.2. Deriving Opening Range Condition for Testing; Finite Element Method Analysis

As crack growth in depth and size over time, and the opening range, will be in accordance with the size of the crack, it is difficult to derive a standardized value for the minimum opening range (mm) for the testing [32]. Therefore, it is important to establish a theoretical basis for the parameter. For the sake of demonstration, this study provides a case study below of a FEM model of a concrete bridge deck (PSC girder type) under the specifications compliant to the Korea Rail Network Authority (KRC standard), but for future application, results derived from modelling must consider case by case design requirements. Typical Korean high-speed railroad operation speed reaches 200~300 km/h, and Korean train dynamic loads average between 75~120 kN/mm^2^ [35]. The deformation load response can produce more realistic data on the performance of waterproofing membranes adhered over a moving crack or joint. For this, a theoretical model is included in the proposed test method only as a demonstration. To propose the opening conditions, the bending moment and stress conditions applied to the waterproofing membrane and the concrete bridge deck due to the train operation dynamic load was analyzed through finite element modelling (FEM).

#### 2.2.1. Finite Element Method Analysis Conditions (Demonstration Using Korean Standards)

For the FEM modelling, a commercial structural analysis program Midas Civil 2019 is used. The length of the analytical model is 10 m, and a 3-dimensional solid element is applied to modeling rail and sleeper. Design properties of the ballasted track are applied to the model. For the boundary condition, the ends of the rail are fixed and the roadbed is considered to have infinite stiffness. The track properties performed in this analysis are compliant to the Korean KRC 14030 standard and are shown in Table 4.

In the analysis, a case of a double-track bridge is modelled and the dynamic responses of only one track are investigated by a time history dynamic load function, and the other track is considered to be the dead load of the bridge as the flexural rigidity of the bridge is usually thousands of times greater than that of the rails. Refer to Figure 4 for the 3-dimensional solid element model of the railroad bridge track.

For the application of the train dynamic load, the train load is depicted in a triangular form as shown in Figure 5, and a time history function analysis method is used in which the load is applied at regular intervals in accordance with the set speed. The load application is made to run at a constant speed over time according to the axial arrangement of the KTX. The proposed 3D coupling element ultimately consists of several rail elements of equal lengths (including the left and right rail), a bridge element, a few sleepers, a series of fasteners, and a series of discrete ballasts. It can also include a bearing that connects a pier node at a supporting point of the bridge. The rails, bridges, and piers are modeled as uniform beams, while each sleeper is modeled as a rigid body, and the lateral and vertical elasticity and damping properties of the fastener, ballast, and bearing are modeled as springs and dampers. Figure 5 illustrates the time history function of the load application on each node of the model, where time difference between *t*_1_ and *t*_2_ is determined according to the train speed and the distance between nodes, and Figure 6 shows the dynamic load results based on the time history graph function.

The above finite element analysis is based on the dynamic load of a KTX train (standard design specification of KTX train: static wheel load 85 kN, 20 cars with 46 wheels) [35] model applied on a single node on the rail model (refer to Figure 7). The dynamic load is applied to one track (right side of the railway bridge track model).

#### 2.2.2. Opening Conditions for Basis of Testing Condition (Demonstration for Obtaining Minimum Opening Range)

Figure 8a shows the FEM analysis of the maximum displacement (displacement perpendicular to the cross section of the bridge structure, in the direction of the dynamic load motion), and the displacement values are derived through deformation analysis based on the time history function of the train wheel load. The red section highlights where the maximum displacement will occur (due to the MIDAS program settings, the red section does not indicate tension stress, but rather highest positive displacement). Under the given conditions and specifications, the results indicate that the bridge deck will undergo a maximum of approximately 1.38 mm displacement. This value can be used as an indicator for this sample study that under the estimated situation where cracks or joints are not present, waterproofing membranes will need to resist a minimum of 1.38 mm displacement. However, for waterproofing membranes, it is pertinent to have the sufficient performance to resist against higher ranges of the displacement, especially in sections with joints or cracks in the bridge structure. Therefore, during testing, waterproofing specimens should not be tested with only the minimum required displacement range setting, but with higher displacement ranges as well. In the coming sections that demonstrate the test method, the minimum opening range for the testing is set to 1.5 mm (for simplification purposes), and the subsequent ranges will increase by the multiples of this value. However, future testing can and should be established based on more realistic settings, as openings in actual concrete structures were significantly larger. Refer to Figure 8a for the deformation analysis results of the track bridge structure model, and Figure 8b for the concrete bridge deck displacement analysis result.

## 3. Proposed Evaluation Method and Criteria for Joint-Opening Resistance Evaluation of Waterproofing Membranes

### 3.1. Test Specimen for Joint-Opening Evaluation

In order to conduct a joint-opening evaluation method, a specimen has to be constructed such that a waterproofing membrane can be installed over a set of concrete/mortar substrate slabs with an artificial crack or joint. As cracks are difficult to simulate with consistent depth and width variables, for this testing demonstration, the opening simulation condition was set compliant to joint conditions only. For the testing, the specimen comprised upper and lower mortar substrate parts (Refer to Figure 9a). The two substrates were placed together at the cross section interface, wherein forming a concrete joint (Refer to Figure 9b). The waterproofing membrane was installed over the substrate surface, completely covering the concrete joint (refer to Figure 9c).

The substrate parts (mortar) of Figure 9a are mixed at a water to cement to sand ratio of 0.4:1:3, during the mortar casting (note, the substrate part mixture ratio can comply to any required specifications or national standards based on the requirement of the testing, and the following procedure will be based on a demonstration version referencing the Korean National Standard (KS) specification for mortar specimen mixture ratio). Threaded conduit parts are placed in their corresponding substrate parts which will be used for connection to the testing device. During casting in the molds, rod tamping is conducted to remove air voids. The freshly cast mortar is cured in a standard laboratory setting for 3 days in ambient conditions (temperature of 20 ± 3 °C and relative humidity of 60 ± 3%). The threaded conduit installed at the lower mortar substrate serves two functions where the part is used to connect to the Universal Testing Machine (UTM) device for testing, and acts as an outlet for leakage during joint opening testing. In this regard, the point of leakage occurrence can be checked immediately during joint opening (refer to Figure 9d). Refer to Figure 9 for details.

### 3.2. Selection and Installation of Waterproofing Systems for Testing

For this testing, 4 types of waterproofing systems were selected: (1) cementitious slurry coating (CSC) in cementitious membrane system, (2) polyurethane spray coating (PUC) in liquid applied membrane system, (3) self-adhesive asphalt sheet (SAS) in asphalt sheet system, and (4) composite asphalt sheet (CAS) in asphalt sheet system. Waterproofing systems are all compliant to the material specifications under KS F 4917 and KS F 4934. Refer to Table 5 for details and illustration of waterproofing layers used in this study.

The waterproofing materials’ (membranes) installation were conducted by a representative of manufacturers to ensure workmanship compliant to the specification. For each waterproofing system, 5 specimens were prepared. The installation was conducted in a laboratory setting with ambient conditions (temperature of 20 ± 3 °C, relative humidity of 60 ± 5%). Once the assembly was completed, the specimens were set to rest for the duration specified by the manufacturers. Refer to Table 6 for the illustrated images of the 4 waterproofing systems’ installation process.

For the cementitious slurry coating (CSC), the non-woven fabric sheet was installed on the mortar substrates, and a fabric layer is impregnated with the cementitious slurry coating material with 2~3 mm thickness. The installed specimens were allowed to cure in accordance with the manufacturer specifications.

For the polyurethane spray coating (PUC), the non-woven fabric sheet was installed onto the mortar substrates, and a fabric layer was impregnated with the polyurethane spray coating material with 2~3 mm thickness. The installed specimens are allowed to cure in accordance with the manufacturer’s specifications.

For the self-adhesive asphalt sheet (SAS), the membrane was cut into a 650 by 150 mm rectangular piece. The membrane was installed on the mortar substrates placed together with the short dimension applied perpendicular to the joint gap. When applying the waterproofing membrane sheets, an overlap joint with a minimum width of 30~50 mm was made.

For the composite asphalt sheet (CAS), the composite membrane was first installed on the mortar substrates with a minimum thickness of about 1~2 mm. The sheet component was also cut into a 650 by 150 mm rectangular piece. The membrane was installed on the mortar substrates placed together with the short dimension applied perpendicular to the joint gap. When applying the waterproofing membrane sheets, an overlap joint with a minimum width of 30~50 mm was made.

### 3.3. Joint-Opening Range for Testing (Sample)

While the analysis data can approximate the minimal opening conditions based on the required conditions, crack or joint opening can reach higher ranges depending on the size of the joint width, and the waterproofing membrane should be able to withstand the deformation conditions at any given realistic range. The minimum opening load width range was set by taking into account the maximum displacement range derived from the FEM modelling sample above in Section 2.2.2, and the rest of the opening ranges were derived by the multiples of the initial opening range of 1.5 mm. Refer to Table 7 for the joint opening ranges.

### 3.4. Testing Apparatus Design and Specimens Setting for Joint-Opening Evaluation

The apparatus consisted of a joint opening simulation chamber that could automatically fill the chamber with water during joint-opening testing. The waterproofing membrane specimen was first secured in the water chamber (or apparatus) by the threaded conduit, which was then filled with approximately 10~15 L of water such that the specimen was completely submerged in water, and inserted into the water chamber for opening load testing. Once installed, the upper substrate was subject to vertical tensile motion in relation to the bottom substrate fixed to the apparatus, thereby simulating a joint-opening stress by 4 opening ranges (refer to Table 7) on the installed waterproofing specimen. Refer to Figure 10 for details.

The joint-opening speed (construction joint-opening rate) was set to 50 mm/min. The final evaluation for the joint opening testing for each specimen was determined by the total number of openings (opening = 1 complete motion of vertical up and down opening) resisted until leakage occurred. Leakage occurrence could be detected and alerted via the leakage outlet installed at the bottom of the joint-opening testing apparatus (as shown in Figure 10c).

## 4. Evaluation Results of Waterproofing Systems Resistance Performance to Joint Opening (Demonstration)

### 4.1. Test Demonstration Result

The evaluation results of 5 specimens’ response to 4 width ranges (1.5, 3.0, 4.5 and 6.0 mm) of joint opening for the 4 respective waterproofing systems (CSC, PUC, SAS and CAS). The results are displayed in Figure 11 which portrays that the performance of the waterproofing membranes changes as the opening width increases for CSC and PUC types, but the changes are barely noticeable for SAS and CAS types. The results here demonstrate that a general trend is clearly delineated in that the resistance performance to joint opening (shown by how many opening cycles each waterproofing system types can resist until leakage occurs) can change as the opening range increases depending on the property of the waterproofing membrane types.

The average opening resistance performance by the number of openings resisted results from the 5 specimens (results from Figure 11) were derived and are shown in Figure 12. For the minimal opening range criteria (1.5 mm), the 4 waterproofing system types were able to display a range from 391 to 693 cycles of resistance performance (number of opening cycles resisted). However, from 3.0 mm onwards, performance range started to differ significantly in that the CSC failed at around below about the 92 opening cycles range, but the other types of waterproofing systems were able to resist at a consistent range (431 to 617 cycles). At 4.5 and 6.0 mm, the average resisted opening cycle for the CSC and PUC was from 7 to 34 cycles, respectively. With the SAS and CAS, the average resisted range of opening cycles remained at a range from 410 to 701 cycles respectively and was not as highly affected by the increasing opening width as was the case for CSC and PUC types.

### 4.2. Grading Criteria and Application Method of the Evaluation Result

As demonstrated above, the number of cycles resisted during the joint-opening testing can become an indicator of joint-opening resistance performance, provided that the proper basis of the opening range conditions has been properly established. As long as the waterproofing membrane is able to prevent leakage through a high range of openings for an extended period of time, this property can be measured and compared with other waterproofing membrane types tested with the same evaluation method. Based on the initial parameters, the results can be compared to a grading system as well for data collection and reference purposes. Table 8 provides a basic example of a grading criteria system based on the above demonstration testing results, and it is proposed that future applications of this evaluation method adopt a similar system, albeit based on the necessary national standard, code or practice.

The demonstration of the evaluation method shows a general trend of decreasing performance as the opening range increases for each specimen type, and a more careful selection process of the waterproofing membrane should be conducted. In the proposed example grading system, only the opening width range is considered as part of the evaluation criteria as a correlative analysis on the maximum number of opening cycles resisted, and the joint resistance performance index has not yet been made clear. One cycle can represent a durability duration factor of 1 day or 1 year depending on requirements of the bridge structure. Nevertheless, the proposed evaluation regime is a step towards an improvement on the practical assessment of waterproofing material performance, because the existing evaluation methods for waterproofing membranes has not been able to approximate the waterproofing performance with regard to joint or concrete opening resistance. Given enough accumulated data on the various different waterproofing membranes, and other degradation factors such as thermal variation-induced concrete expansion/contraction, chemical exposure, direction of tension, etc., it is expected that a selection of appropriate waterproofing membranes that improve the sustainability of railway bridge structures can be possible. The modelling example, the evaluation method demonstration, and results and the grading system are a very basic format of the proposed idea, but still establishes the requirement for more future application of these types of evaluation method.

## 5. Conclusions

In this study, a joint-opening resistance evaluation method of different waterproofing systems (membranes or materials) is proposed, with the results of which a waterproofing system can be selected that complies with the joint-opening degradation conditions of high-speed railroad bridge decks. The study offers the following conclusions:(1)The opening range to be used for the testing was determined based on the FEM analysis of a concrete bridge deck deformation due to the wheel load of train operation. A FEM modelling of a railway bridge track based on the conditions compliant to the KRC 14030 was used for the demonstration of the proposed evaluation method, and future application of this test method, if applicable, can adopt a similar regime to derive the minimum opening range for the testing.(2)The demonstration of this evaluation method serves to clearly delineate the difference in the joint-opening resistance performance of waterproofing systems commonly used in railroad bridge decks. The proposed joint opening resistance performance testing in this study evaluated the respective performance of 4 different waterproofing membranes (CSC, PUC, SAS, CAS) based on changing joint opening width. The demonstration was able to determine that the CAS type has the highest relative joint-opening resistance performance among the 4 types.(3)Further research must be performed to improve on the accuracy of the evaluation method such that it can be made employable in different national standards. The research results in this study are intended to assist in the development of laboratory watertightness testing methods for waterproofing membranes subjected to various degradation conditions (one such being resistance to joint opening). The demonstrated evaluation regime will have to include more accurate variables that comply with the environmental conditions of railroad bridge decks. When this can be accomplished through further research, selection of appropriate waterproofing membranes with high watertight performance can be selected for high maintenance infrastructures such as railway bridges. It can also provide a platform for developing higher-performance waterproofing membranes through competition based on the evaluation results. It is hoped that future designs of railroad bridge waterproofing can take into consideration the effect of waterproofing on the sustainability of the structure against water damage-based corrosion of PSC, and that the proposed evaluation can be a contributing factor in the selection of optimal waterproofing systems.

## Figures and Tables

**Figure 1 materials-13-04229-f001:**
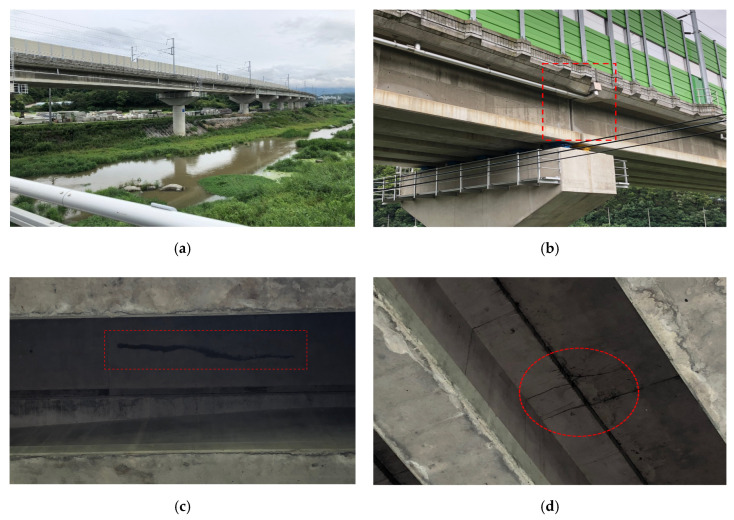
Bridge structure degradation status; (**a**) bridge overview, (**b**) bridge joint section, (**c**) bridge underside leakage crack 1, (**d**) bridge underside leakage crack 2.

**Figure 2 materials-13-04229-f002:**
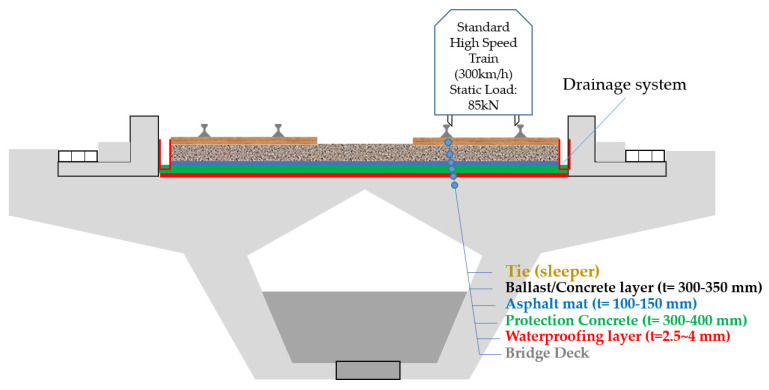
Sample pre-stressed concrete (PSC) track bed structure design layout with waterproofing system.

**Figure 3 materials-13-04229-f003:**
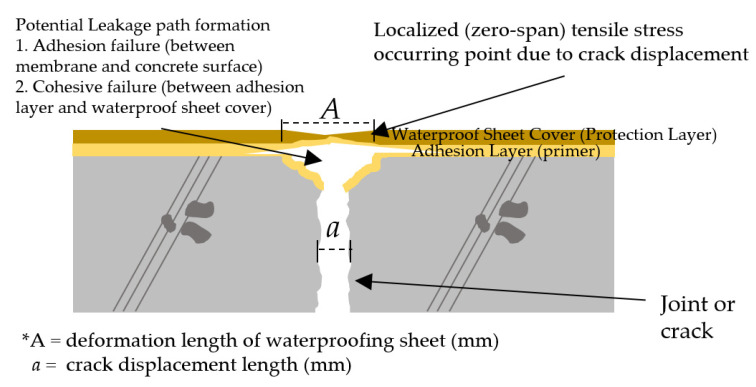
Zero-span tensile stress on waterproofing membrane due to joint or crack opening.

**Figure 4 materials-13-04229-f004:**
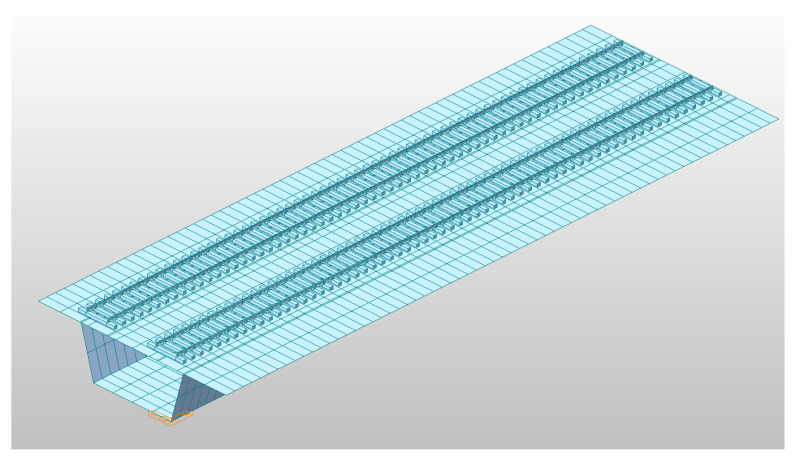
Three-dimensional analytical model (L = 10 m).

**Figure 5 materials-13-04229-f005:**
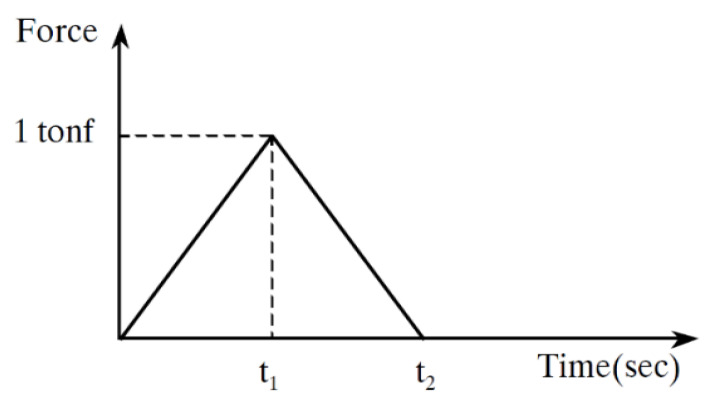
Time histogram function concept.

**Figure 6 materials-13-04229-f006:**
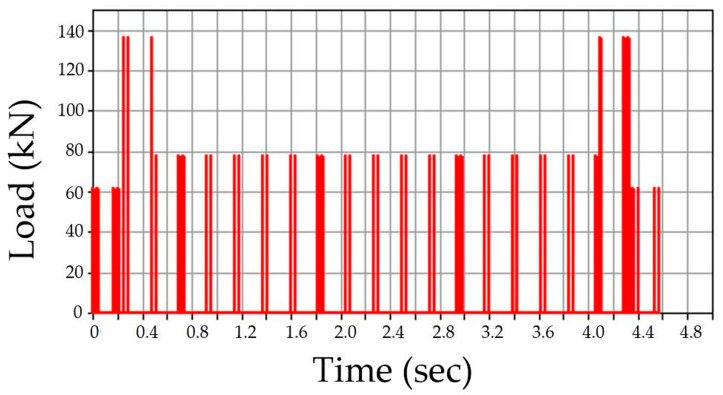
Time histogram function results for the finite element analysis (kN).

**Figure 7 materials-13-04229-f007:**
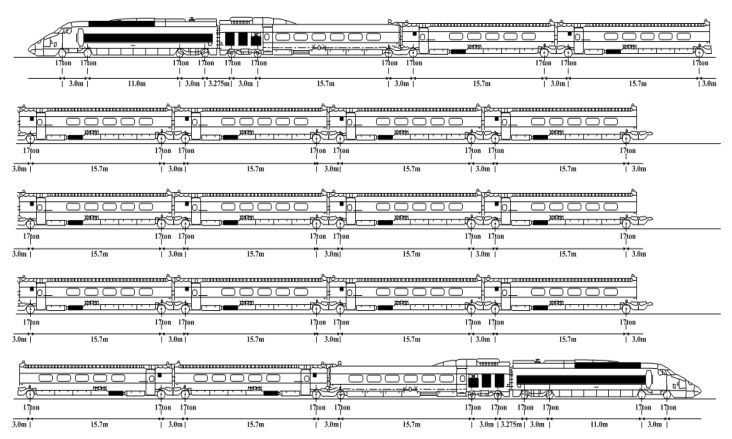
Korea Train Express (KTX) wheel composition [35].

**Figure 8 materials-13-04229-f008:**
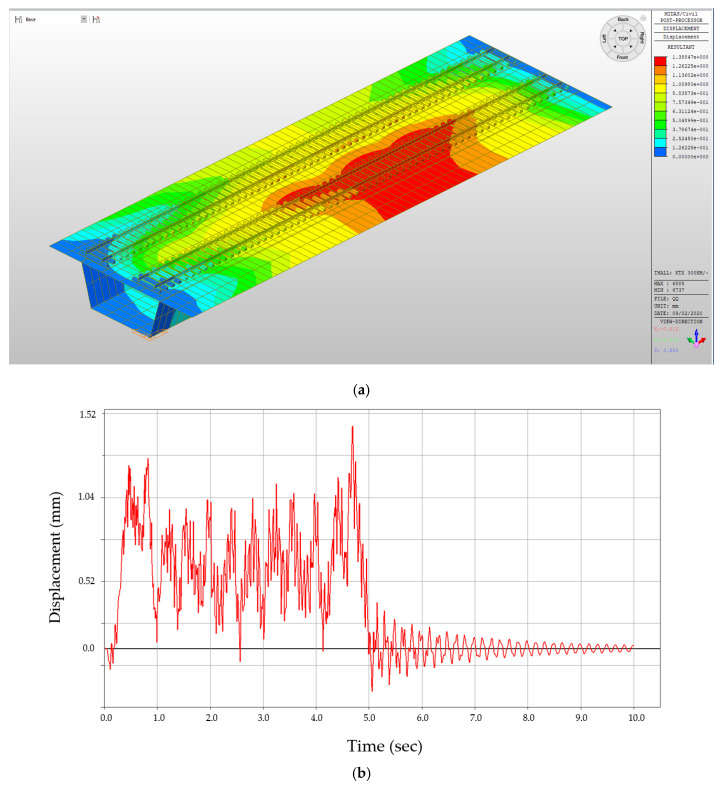
FEM analysis result (sample); (**a**) deformation analysis results of the track bridge structure model, (**b**) concrete bridge deck displacement analysis (1.38 mm).

**Figure 9 materials-13-04229-f009:**
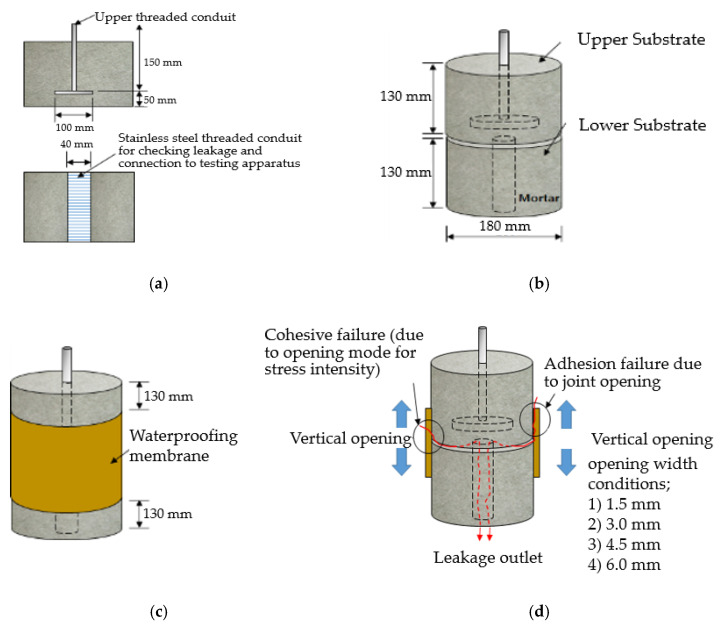
Crack opening testing specimen (mortar substrate) structure layout; (**a**) mortar substrate inner structure schematic, (**b**) mortar substrate dimensions and labels, (**c**) waterproofing membrane installed (henceforth called specimen), (**d**) opening resistance testing illustrated.

**Figure 10 materials-13-04229-f010:**
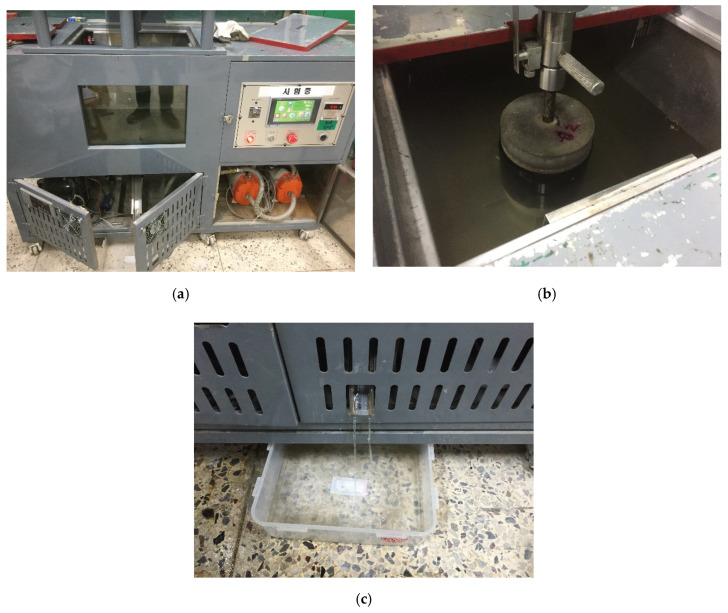
Testing apparatus illustrated. (**a**) Overview of the joint opening simulation water chamber; (**b**) specimen installed in the joint opening simulation water chamber; (**c**) leakage outlet for alerting leakage occurrence during testing.

**Figure 11 materials-13-04229-f011:**
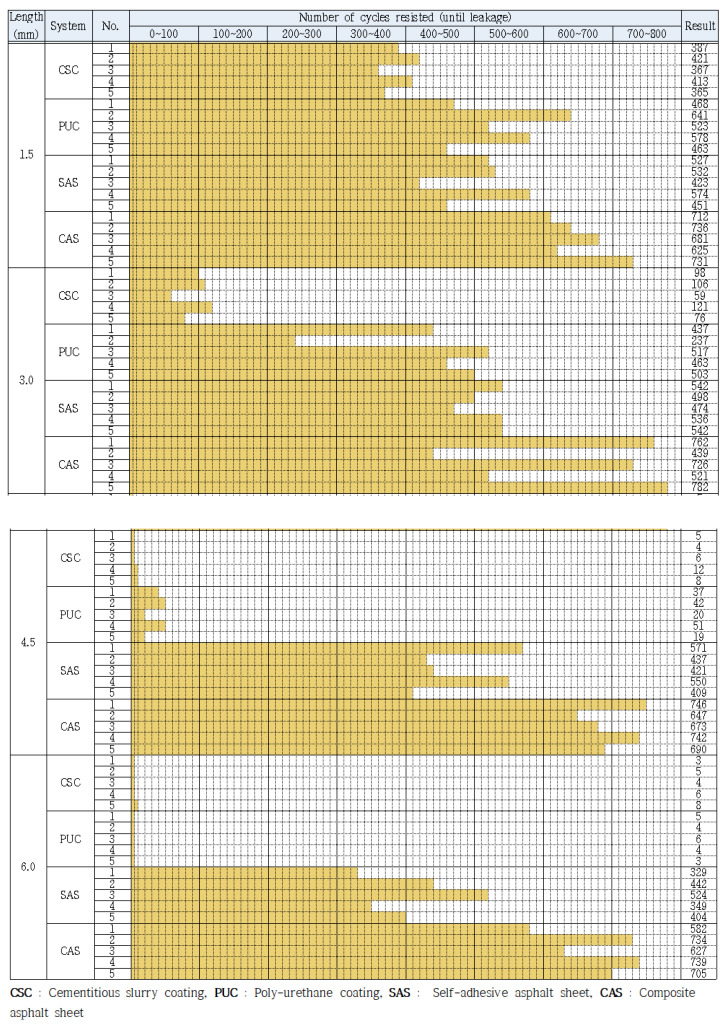
Joint opening resistance performance result (per opening range).

**Figure 12 materials-13-04229-f012:**
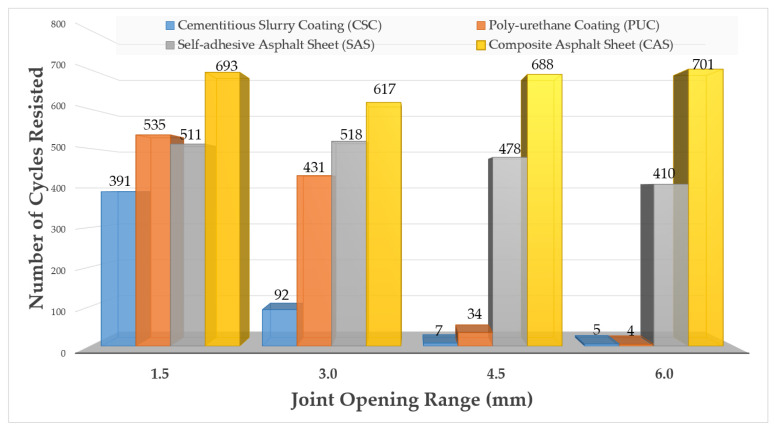
Joint opening resistance performance result of the respective waterproofing system types (per opening range).

**Table 1 materials-13-04229-t001:** Representative waterproofing systems of railroad bridges decks.

Methods	Sheet Membrane System (e.g., Asphalt Sheet, Composite Sheet)	Liquid Applied Membrane System (e.g., Polyurethane Coating)	Cementitious Membrane System (e.g., Cementitious Slurry Coating)
	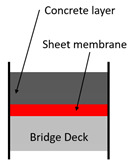	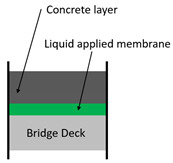	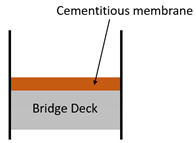
Thickness (mm)	2~4	2~3	2~3
Average modulus of elasticity (N/mm^2^)	8.72 GPa~12.51 GPa	0.93 GPa~5.31 GPa	10.14 GPA~14.38 GPA
Protection layer type and thickness (mm)	Asphalt concrete: 40	Asphalt concrete: 40	None

**Table 2 materials-13-04229-t002:** Representative classification of waterproofing systems for railroad bridge deck in international application.

Nations	Waterproofing Materials and Types		
	Sheet Membrane System	Liquid Applied Membrane System	Cementitious Membrane System
U.S.	O	O	X
European Nations	O	O	X
Japan	O	O	X
Korea	O	O	O
China	O	O	O

**Table 3 materials-13-04229-t003:** Standards related to waterproofing membrane performances.

Performance Criteria	National Standards				
	ASTM D 7832	KS F 4917, KS F 4935	GB 50108	BS 8102	JIS A 6909, JIS A 6021, JIS A 6008
Tensile strength	O	O	O	O	O
Elongation	O	O	O	O	O
Chemical resistance	O	O	O	O	O
Adhesion strength	O	O	O	O	O
Crack bridging *	O	O	X	X	X
Joint opening resistance	X	X	X	X	X

* Conventional crack bridging test found in ASTM relies on visual observation of the waterproofing membrane after joint/crack opening cycling. Assessing whether the waterproofing membrane retains waterproofing performance after opening cycling is not possible with this test method.

**Table 4 materials-13-04229-t004:** Railway bridge design condition for finite element method (FEM) analysis.

Division	Element	Item	Input Data
Rail	3-dimensional solid element	Type	UIC60
		Modulus of elasticity (kN/mm^2^)	206
		Weight density (kN/m^3^)	0.785
		Poisson’s ratio	0.30
		Height (mm)	172
		Width of rail head (mm)	72
		Width of rail base (mm)	150
		Area (mm^2^)	76.86
		X axis moment of inertia (mm^4^)	3.05 × 10^−6^
		Z axis moment of inertia (mm^4^)	5.13 × 10^−6^
		Tensile strength (N/mm^2^)	880
Pre-cast concrete sleeper	3-dimensional solid element	Modulus of elasticity (kN/mm^2^)	33.5
		Weight density (kN/m^3^)	1.3
		Poisson’s ratio	0.18
		Width (mm)	265
		Length (mm)	2400
		Height (mm)	195
		Z axis moment of inertia (mm^4^)	1.41 × 108
Rail pad	3-dimensional spring-damper element	Vertical direction modulus of elasticity, *k_w_* (kN/mm)	100
		Vertical direction damper coefficient (kN sec/mm)	0.098
Ballast	3-dimensional spring-damper element	Vertical direction modulus of elasticity, (*k_w_* (kN/mm)	200
		Vertical direction damper coefficient (kN sec/mm)	0.98

**Table 5 materials-13-04229-t005:** Types of waterproofing membranes evaluated.

**Cementitious Slurry Coating (CSC)**	**Polyurethane Spray Coating (PUC)**
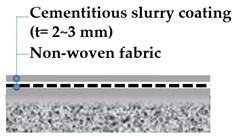	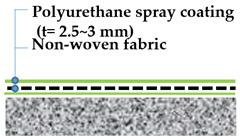
**Self-Adhesive Asphalt Sheet** **(SAS)**	**Composite Asphalt Sheet** **(CAS)**
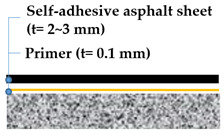	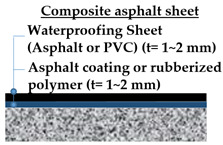

**Table 6 materials-13-04229-t006:** Four waterproofing system specimens’ installation process.

Specimen	Procedure			
	Substrate Preparation	Base Preparation	Membrane Installation	Specimen Complete
Cementitious slurry coating	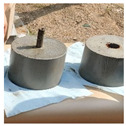	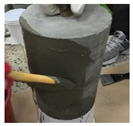	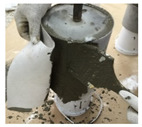	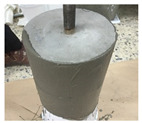
Polyurethane spray coating	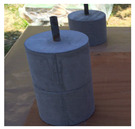	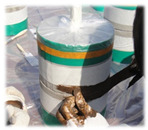	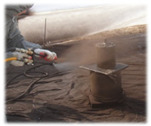	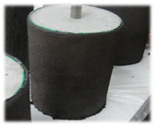
Self-adhesive asphalt sheet	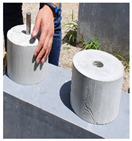	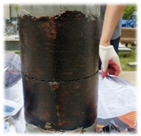	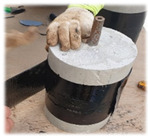	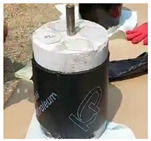
Composite asphalt sheet	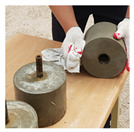	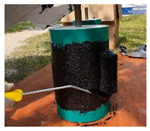	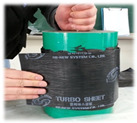	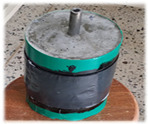

**Table 7 materials-13-04229-t007:** Opening range for testing demonstration.

Opening Range	Description (Example Criteria)
1.5 mm ^1^	Normal deformation range of normal PSC substrate of 10 m span when no cracks are present or crack opening of 1.5 mm is expected.
3.0 mm	Average joint opening range of normal PSC substrate of 10 m span. Joints of at least 3.0 mm width will be subject to this range of opening when the bending moment due to train wheel load is considered.
4.5 mm	High joint opening range of normal PSC substrate of 10 m span. Joints of at least 4.5 mm width depth will be subject to this range of opening when the bending moment due to train wheel load is considered.
6.0 mm	Extreme joint opening range of normal PSC substrate of 10 m span. Joints of this will be subject to this range of opening when the bending moment due to train wheel load is considered.

^1^ Ranges and description may vary in accordance to the required testing criteria and the design specification of the bridge structure and environmental condition.

**Table 8 materials-13-04229-t008:** Grading criteria (example).

Grade	Description (Example)
Minimal opening resistance capacity grade: (e.g., 1.5 mm opening resisted on average)	Materials such as **cementitious slurry system**, capable of handling up to 1.5 mm opening range but no higher to ensure long term durability. Materials of this grade can be used in: low expectation of environmental degradation factors; cracks will not occur. Applicable material types: CSC, PUC, SAS, CAS
Moderate opening resistance capacity grade (e.g., 3.0 mm opening resisted on average)	Materials such as polyurethane coating, capable of handling up to 3.0 mm opening range but no higher. Materials of this grade can be used in: low expectation of environmental degradation factors; minor cracks may occur. Applicable material types: PUC, SAS, CAS
High opening resistance capacity grade (e.g., 4.5 mm opening resisted on average)	Materials such as self-adhesive sheets, capable of handling up to 4.5 mm opening range. Materials of this grade can be used when: environmental degradation factors can occur; cracks may occur naturally. Applicable material types: SAS, CAS
Excellent opening resistance capacity grade (e.g., 6.0 mm opening resisted on average)	Materials such as self-adhesive sheets, capable of handling up to 4.5 mm opening range. Materials of this grade can be used when: environmental degradation factors can occur; extreme cracking may occur naturally Applicable material types: CAS

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
