# Peer review of "Joint- or Crack-Opening Resistance Evaluation of Waterproofing Material and System for Structural Sustainability in Railroad Bridge Deck"

_materials, 2020, doi:10.3390/ma13194229_

Round 1

Reviewer 1 Report

Comments

This study experimentally investigated the crack displacement resistance evaluation of waterproofing material. The conclusion section does not represent any major findings, rather it is the part of methodology. 

Author Response

The authors would like to express their sincerest gratitude for taking time out of the reviewer’s busy schedule to read the article and provide their valued advice, revision points and comments. As there were numerous points of requested revision by different reviewers, the article has gone through some extensive change since the last version. In the below, the authors have provided as detailed as possible response to the respective comments.

Reviewer 1

Comment 1

This study experimentally investigated the crack displacement resistance evaluation of waterproofing material. The conclusion section does not represent any major findings, rather it is the part of methodology. 

Response 1

1) The purpose of the study of this paper was not to simply experimentally investigate the joint or crack displacement resistance performance of waterproofing materials for railway bridge deck structures., but to propose a possible method that makes this evaluation possible, and demonstrate this method. While the variables (types of waterproofing membranes tested, displacement range, method to derive the displacement range parameters) are interchangeable in accordance the requirements of different national standards (as the bridge structure scale determines the expected range of displacement, the modelling parameters are expected to change case by case).

2) The main purpose of the conclusion section is to outline that the evaluation method can be used to clearly outline that the tested waterproofing membrane types will have different performance under the same displacement range condition. The conclusion section as well as the introduction and the results have been revised to state the purpose of the paper more clearly (please refer to the revised Introduction section lines 62-94, and the Results (lines 442-470) and Conclusion Sections (lines 478-497).

3) The paper attempts explain the concept on the effect of zero-span tensile stress on the adhered waterproofing membrane on the concrete railway bridge deck surface in the situation where a crack or joint is present. As is commonly known by the concept of stress concentration factor, when stress is applied to a concrete member, the stress is concentrated at the tip of the crack. When the crack displaces (due to the bending moment of the bridge caused by the wheel load, thermal expansion contraction, or other environmental causes), the localized stress will cause the adhered waterproofing membrane to receive higher stress as well (source: Kunieda, M. Rokugo, K. Recent Progress of HPFRCC in Japan—Required Performance and Applications. J. Adv. Concr. Technol. 2006, 4, 19–33). The common expression to describe the waterproofing membrane’s performance to resist against this stress is called “crack bridging” property (ASTM C 1305 Standard Test Method for Crack Bridging Ability of Liquid-Applied Waterproofing Membrane).

4) It is important to highlight this section as inability to have crack bridging property for waterproofing membranes in railway bridge structures where cracked sections displace constantly will break or allow the formation of leakage path due to cohesive (break within the waterproofing membrane layer) or adhesive (peel off from concrete surface) failure. To overcome this problem, a proper waterproofing membrane evaluation method needs to be developed and demonstrated to show that it can be used for selection prior to use in concrete railway bridge decks in this paper).

Lastly, the paper has been significantly revised since the last version due to the careful review and comments from the other reviewers, and the authors hope that the above points of revision based on the reviewers’ comments are sufficient for reconsideration, and your generous re-review of the paper. Thank you kindly again for taking the time out of your busy schedule to review this paper.

Reviewer 2 Report

This manuscript deals with Joint or Crack Displacement Resistance Evaluation of Waterproofing Material and System for Structural Sustainability in Railroad Bridge Deck.
This is an interesting topic and research.
The manuscript has potentially new knowledge.
The manuscript presents a comprehensive approach to this issue.

The manuscript has the usual structure.
However, the manuscript does not have part of the discussion.

The chosen methods are suitable.

However, the description of the methods is not sufficient. In particular the field of numerical modeling - FEM.

The current state of the issue - Introduction - is insufficiently described. Current research in the research field is significantly larger and there are significantly more sources for comparison and discussion.

This part needs to be fundamentally reworked.

Research assumptions, research methods, results, and conclusions are represented.

The presentation of research and results is insufficient.

Table 3 is on two pages.
The lines 183-186 - of the sentence are not properly worded.
Figure 3. - Images should be enlarged. Scales cannot be distinguished.
Table 6 is on two pages.
Table 9 has the wrong format according to the template.
Figure 9 is on two pages.

With regret, I have to say that the authors have not put their, enough best effort into an article and presentation of research and result. Due to the above reasons, I must recommend to not accept the paper for publication in the present form.
In summary, it can be still manuscript significantly improved for the reader. I recommend the major revision of the manuscript.

Author Response

The authors would like to express their sincerest gratitude for taking time out of the reviewer’s busy schedule to read the article and provide their valued advice, revision points and comments. As there were numerous points of requested revision by different reviewers, the article has gone through some extensive change since the last version. In the below, the authors have provided as detailed as possible response to the respective comments;

Reviewer 2

This manuscript deals with Joint or Crack Displacement Resistance Evaluation of Waterproofing Material and System for Structural Sustainability in Railroad Bridge Deck. This is an interesting topic and research. The manuscript has potentially new knowledge. The manuscript presents a comprehensive approach to this issue.

Comment 1

The manuscript has the usual structure. However, the manuscript does not have part of the discussion.

Response 1

The title of section 2 has been revised to more clearly state where the discussion section for the paper structure begins. (Please refer to Section 2 Theoretical discussion (line 195) in the revised paper for details) The chosen methods are suitable.

Comment 2

However, the description of the methods is not sufficient. In particular the field of numerical modeling - FEM.

Response 2

More details on the FEM modelling has been included (the modelling parameters have been revised as well). The design parameters, time history function for the dynamic loading conditions, the train (KTX) wheel load specification has been included. Please refer to the revised sections 2.2.1 (Table 4 / Figure 4, 5, 6 and 7) to 2.2.2(Figure 8 a), b)) for details.

Comment 3

The current state of the issue - Introduction - is insufficiently described. Current research in the research field is significantly larger and there are significantly more sources for comparison and discussion. This part needs to be fundamentally reworked.

Response 3

1) The introduction section has been revised to include more sources in the research field. However, please keep in mind that while the field of railway bridge maintenance and sustainability may be a large field, the specific field of waterproofing material and waterproofing construction in railway bridge is not as large (at least as far as Korean, Chinese, Japanese, and English language sources are concerned). Existing sources do exist in that the references, specifications, guidelines and standards mention the requirement for waterproofing, but specificities are lacking. Just upon referencing the AREMA (American Railway Engineering and Maintenance-of-Way Association) codes and practices, under Section 29.2 Waterproofing and onwards, (https://www.yumpu.com/en/document/read/51750577/arema-part-29-waterproofing), one can easily denote that the referenced standards are purely waterproofing material testing methods (ASTM standards), and methods described are vague, using obscure terms such as “need to be adequately protected,” “durable and effective,” “essential” are used to describe the waterproofing construction process.

2) These trends can be found in most conventional references regarding waterproofing. This is due to the fact that while the theoretical application of waterproofing evaluation has been clearly established (references talking about this point has been included in the revised version, please refer to the revised Introduction Section of the paper), a practical solution and evaluation method has not yet been developed.      

3) Standards such as the GB 50108 (Below-grade structure waterproofing) could be useful for structural waterproofing as they provide specific and quantifiable grading systems for waterproofing membranes, but for railway bridge deck waterproofing where the loading and shear stress conditions are fundamentally different, references that include a laboratory method that utilizes joint or crack displacement simulation for testing waterproofing material performance is difficult to find. This point underlines the purpose of this paper, which is to establish the necessity of this evaluation method and a demonstration of this method for hopes of future application in this field. 

Comment 4

Research assumptions, research methods, results, and conclusions are represented. The presentation of research and results is insufficient.

Response 4

1) As is mentioned briefly in the response of the above comment, the purpose of the study of this paper was not to simply experimentally investigate the crack displacement resistance performance of waterproofing materials, but to propose a possible method that makes this evaluation possible, and demonstrate this method. While the variables (types of waterproofing membranes tested, displacement range, method to derive the displacement range parameters) are interchangeable in accordance the requirements of different national standards (as the bridge structure scale determines the expected range of displacement, the modelling parameters are expected to change case by case).

2) The main purpose of the conclusion section is to outline that the evaluation method can be used to clearly outline that the tested waterproofing membrane types will have different performance under the same displacement range condition. The conclusion section as well as the introduction and the results have been revised to state the purpose of the paper more clearly.

Comment 5

Table 3 is on two pages. The lines 183-186 - of the sentence are not properly worded.
Figure 3. - Images should be enlarged. Scales cannot be distinguished.
Table 6 is on two pages.
Table 9 has the wrong format according to the template.
Figure 9 is on two pages.

Response 5

- Table 3 has been revised to be in 1 page

- The lines 183-186 has been revised

- Figure 3.  has been revised such that the scales (displacement (x direction)) should now be visible (due to the format of the Midas program, the scale cannot be isolated or be extracted

- Table 6 has been revised to be in 1 page

- Table 9 has been removed to simply and make the results section clearer

- Figure 9 has been removed.

(Tables and Figures being in two pages can be edited at the later stages)

Comment 6

With regret, I have to say that the authors have not put their, enough best effort into an article and presentation of research and result. Due to the above reasons, I must recommend to not accept the paper for publication in the present form. In summary, it can be still manuscript significantly improved for the reader. I recommend the major revision of the manuscript.

Response 6

Lastly, the paper has been significantly revised since the last version due to the careful review and comments from the other reviewers, and the authors hope that the above points of revision based on the reviewers’ comments are sufficient for reconsideration, and your generous re-review of the paper. Thank you kindly again for taking the time out of your busy schedule to review this paper.

Reviewer 3 Report

Materials-903292 Review (v1)

I commend the authors on their study about the evaluation method for the waterproofing material and system used in the railroad bridge deck. Some meaningful tests were designed and carried out according to the FEM analysis result of the joint or crack displacement range. The work makes some contributions towards the future selection of waterproofing materials for the sustainability of high-speed railroad bridge deck structures. Upon a review, I have the following comments below.

  1. The full text is not concise enough, especially in presenting the test results and conclusions.
  2. In the section of 2.1, the expression of “… the joints/cracks are pathways for chloride ions mixed with water to penetrate concrete matrix and cause carbonation of concrete. The reaction of Ca(OH)2 and chloride ions to form CaCO3 lowers the pH of concrete.” is incorrect. It is not chloride ions, but carbon dioxide mixed in water that causes the carbonation of the concrete.
  3. Line 106~107, “t” should be “thickness”, expressions should be complete. Line 108, capitalize the first letter of a sentence. Line 116, use a period at the end of the sentence. The whole text should be checked and reviewed. (Line 149, 212, 253, et al)
  4. Line 132~135, why are there two maximum stress values generated at the waterproofing membrane under the action of thermal loading? Are they corresponding to different working conditions? Give the necessary statement.
  5. Line 135~136, the logic is not very clear. Are you trying to express that the conventional FEM modelling only considers one of two kinds of loads? If so, the statement is confusing. “However, conventional FEM modelling conditions only consider either the load from the rolling stock or the thermal load on the concrete deck.”
  6. Line 151, it is suggested to replace K with °C in the units of the expansion coefficient α and temperature difference Δt. Be consistent with other expressions.
  7. Line 155~157, why is the temperature difference between day time and night (20 °C) instead of the seasonal maximum temperature difference was used to calculate the thermal expansion? How is the value of 1.176× 10-2 m calculated by equation 1? Please state clearly the values of parameters in equation 1.
  8. Similarly, the lack of parameter values in equations 2 and 3.
  9. Table 4, What’s the meaning of “Bridge sength”, “Gauge sength”?
  10. How to consider the interface characteristics between the waterproofing materials and concrete bridge deck in the model? What are the application method of the cyclic train load and the time history function of the dynamic load? Key modeling parameters and processes should be given.
  11. Line 221, how is the value of 5.415× 10-2 m calculated? Please give specific values of each parameter in equations 2 and 3. Besides, can we extract this value from the simulation results directly?
  12. Figure 3 is not clear, especially the legend inside.
  13. What is the sense of section 2.4? What is the relationship between the following content and this part?
  14. Section 3.1 and Figure 6, please explain how to check the leakage?
  15. Section 4 needs a major revision. The results presented in Table 9, Figure 8, and Figure 9 are exactly the same. Figures 10 and 11 are the same too. The authors should simplify this part and highlight some useful results and conclusions from these tests.

Author Response

The authors would like to express their sincerest gratitude for taking time out of the reviewer’s busy schedule to read the article and provide their valued advice, revision points and comments. As there were numerous points of requested revision by different reviewers, the article has gone through some extensive change since the last version. In the below, the authors have provided as detailed as possible response to the respective comments;

Reviewer 3

I commend the authors on their study about the evaluation method for the waterproofing material and system used in the railroad bridge deck. Some meaningful tests were designed and carried out according to the FEM analysis result of the joint or crack displacement range. The work makes some contributions towards the future selection of waterproofing materials for the sustainability of high-speed railroad bridge deck structures. Upon a review, I have the following comments below.

 The full text is not concise enough, especially in presenting the test results and conclusions.

Comment 1

In the section of 2.1, the expression of “… the joints/cracks are pathways for chloride ions mixed with water to penetrate concrete matrix and cause carbonation of concrete. The reaction of Ca(OH)2 and chloride ions to form CaCO3 lowers the pH of concrete.” is incorrect. It is not chloride ions, but carbon dioxide mixed in water that causes the carbonation of the concrete.

Response 1  

The authors would like to apologize for making such an elementary mistake. The expression has been revised accordingly. Please refer to the revised Section 1.2 lines 115-118 for details

Comment 2

Line 106~107, “t” should be “thickness”, expressions should be complete. Line 108, capitalize the first letter of a sentence. Line 116, use a period at the end of the sentence. The whole text should be checked and reviewed. (Line 149, 212, 253, et al)

Response 2

Expression “t” in lines 136 to 142 (in the revised version of the paper) has been revised such that “t” is now “thickness,” and the rest of the lines have been reviewed and revised accordingly.

Comment 3

Line 132~135, why are there two maximum stress values generated at the waterproofing membrane under the action of thermal loading? Are they corresponding to different working conditions? Give the necessary statement.

Response 3

One of the stress value was supposed to be designated to only wheel load, and another was supposed be including thermal stress + wheel load. However, theoretical application has been revised such the FEM demonstration only uses wheel load to derive the minimum displacement range condition for testing, and sections discussing thermal loading has been removed from the paper. The focus on the stress conditions has been revised to center around bending moment (concrete bridge deck deformation) due to wheel load only. Please refer to Sections 2.2.1 and 2.2.2 in the revised paper for details.

Comment 4

Line 135~136, the logic is not very clear. Are you trying to express that the conventional FEM modelling only considers one of two kinds of loads? If so, the statement is confusing. “However, conventional FEM modelling conditions only consider either the load from the rolling stock or the thermal load on the concrete deck.”

Response 4

The statement was originally supposed to indicate that conventional FEM modelling used in Korea for dynamic stability assessment of railway bridge/track design only considers dynamic and static wheel load (and not thermal or environmentally induced loading). However this section has been revised and is no longer relevant in the paper. Please refer to the sections 2.2 and 2.2.1 (lines 220-241) for details.

Comment 5

Line 151, it is suggested to replace K with °C in the units of the expansion coefficient α and temperature difference Δt. Be consistent with other expressions.

Line 155~157, why is the temperature difference between day time and night (20 °C) instead of the seasonal maximum temperature difference was used to calculate the thermal expansion? How is the value of 1.176× 10-2 m calculated by equation 1? Please state clearly the values of parameters in equation 1.

Response 5

The section involving thermal stress load condition has been removed. While this is important to consider thermal expansion and contraction of the concrete bridge deck and incorporate it into deriving the minimum displacement range, too many variables in this study (where the main purpose is to discuss the necessity for a practical evaluation for joint/crack displacement resistance performance of waterproofing membranes for concrete bridge deck and demonstration) is a cause for confusion for the readers. To simplify and solidify the theoretical process, only the discussion of the mechanical degradation due to wheel load has been included in the paper.

Comment 6

Similarly, the lack of parameter values in equations 2 and 3.

Response 6

Equations 2 and 3 were used to calculate the approximate displacement length of the waterproofing membrane and the concrete deck under thermal load and wheel load. The stress variables (parameter values) to be used in these equations are interchangeable based on the different national standards, practices or specifications (based on the dimensions/scale of the railway bridge deck and the static load of the train), and it was originally the intent of the authors to include these equations as they are commonly used in Korea to calculate the deformation of railway components under wheel load. However, authors changed the scope of the demonstration to obtain the minimum joint displacement range to rely mostly on the FEA modelling, thus equations 2 and 3 are no longer applied in this paper.

Comment 7

Table 4, What’s the meaning of “Bridge sength”, “Gauge sength”?

Response 7

The words were spelling errors. “Sength” has been revised to “Length” (Table 4 has been revised in Section 2.2.1)

Comment 8

How to consider the interface characteristics between the waterproofing materials and concrete bridge deck in the model? What are the application method of the cyclic train load and the time history function of the dynamic load? Key modeling parameters and processes should be given.

Response 8

Originally, the compliant material properties were given to the waterproofing layer in the model (plate model) and an elastic link at the individual nodes were given based on the adhesion strength value of the waterproofing membrane types on to the concrete bridge deck surface model, but as the main purpose of the modelling is to derive the concrete deck displacement (horizontal direction of X and Y) and not of the waterproofing membranes, the authors decided to remove this element from the modelling to simplify the modelling process in the hopes of making the purpose more clear.

Comment 9

Line 221, how is the value of 5.415× 10-2 m calculated? Please give specific values of each parameter in equations 2 and 3. Besides, can we extract this value from the simulation results directly?

Response 9

This value was calculated based on the expected maximum stress conditions applied to the concrete deck when both wheel and thermal load were considered, but as explained in the above, the scope of the technical process for deriving the minimum joint displacement range has been limited to the results obtained from the FEA model only.

Comment 10

Figure 3 is not clear, especially the legend inside.

Response 10

Figure 3.  has (now figure 5 a) in the revised version of the paper) been revised such that the scales (displacement (x direction)) should now be visible (due to the format of the Midas program), the scale cannot be isolated or be extracted. However, in case the legend is still unclear, a graph on the displacement analysis result has been included in Figure 8, b).

Comment 11

What is the sense of section 2.4? What is the relationship between the following content and this part?

Response 11

1) Section 2.4 (now section 2.1) is the theoretical part that describes the necessity of the joint displacement. The section explains the concept on the effect of zero-span tensile stress on the adhered waterproofing membrane on the concrete deck surface in the situation where a crack or joint is present. As is commonly known by the concept of stress concentration factor, when stress is applied to a concrete member, the stress is concentrated at the tip of the crack. When the crack displaces (due to the bending moment of the bridge caused by the wheel load, thermal expansion contraction, or other environmental causes), the localized stress will cause the adhered waterproofing membrane to receive higher stress as well (source: Kunieda, M. Rokugo, K. Recent Progress of HPFRCC in Japan—Required Performance and Applications. J. Adv. Concr. Technol. 2006, 4, 19–33). The common expression to describe the waterproofing membrane’s performance to resist against this stress is called “crack bridging” property (ASTM C 1305 Standard Test Method for Crack Bridging Ability of Liquid-Applied Waterproofing Membrane).

2) It is important to highlight this section as inability to have crack bridging property for waterproofing membranes in railway bridge structures where cracked sections displace constantly will break or allow the formation of leakage path due to cohesive (break within the waterproofing membrane layer) or adhesive (peel off from concrete surface) failure.

Comment 12

Section 3.1 and Figure 6, please explain how to check the leakage?

Response 12

Method for describing how to check the leakage has been included in Section 3.4, Figure 10 c). This is also related to the concept explained in Section 3.1, Figure 9 d).

Comment 13

Section 4 needs a major revision. The results presented in Table 9, Figure 8, and Figure 9 are exactly the same. Figures 10 and 11 are the same too. The authors should simplify this part and highlight some useful results and conclusions from these tests.

Response 13

1) The purpose of the study of this paper was not to simply experimentally investigate the crack displacement resistance performance of waterproofing materials, but to propose a possible method that makes this evaluation possible, and demonstrate this method. While the variables (types of waterproofing membranes tested, displacement range, method to derive the displacement range parameters) are interchangeable in accordance the requirements of different national standards (as the bridge structure scale determines the expected range of displacement, the modelling parameters are expected to change case by case).

2) The main purpose of the conclusion section is to outline that the evaluation method can be used to clearly outline that the tested waterproofing membrane types will have different performance under the same displacement range condition. The conclusion section as well as the introduction and the results have been revised to state the purpose of the paper more clearly. Also, in Section 4, the separate figures 10 and 11 (in the previous version of the paper) and the relevant analysis have been consolidated, and only Figure 11 and Figure 12 have been used for the revised version of the paper. Please refer to the revised Section 4, subsections 4.1 and 4.2 for details.

Also, it should be noted that the sections, particularly the parts about the 2 separate degradation conditions (thermal stress induced expansion and contraction and wheel load induced bending moment), as well as the equations and calculations, have been consolidated to clarify the presentation of the process of using the proposed evaluation method in real situations. The authors agreed that theories included in the previous version of the paper were too many when all that was need was to clarify just one factor that causes the displacement range to increase. Therefore, more focus and explanation on the methodology to derive the displacement range through a FEA modelling on the bending moment was provided.

Lastly, the paper has been significantly revised since the last version due to the careful review and comments from the other reviewers, and the authors hope that the above points of revision based on the reviewers’ comments are sufficient for reconsideration, and your generous re-review of the paper. Thank you kindly again for taking the time out of your busy schedule to review this paper.

Reviewer 4 Report

Authors design and verified the novel procedure of evaluation the waterproofing systems. The paper is written on a reasonable manner, the figures are clear, but the Reviewer has some remarks, which are presented below:

  1. Introduction, lines 48-50: "Numerous other researchesindicate that waterproofing of highway bridges and railroad bridge is a topic of high interest in China, where high performance waterproofing membranes types are regularly tested and reported in the recent research papers" - please add references.

2. The Introduction is very brief. Despite that the Authors clearly define the purpose of the research, in reviewers' opinion the literature review is very poor. Please add the literature about cracks or debondings in concrete structures or bridges.

3. Did Authors considered the concrete as homogeneous and linearly elastic? Concrete is brittle material. The initial microcracks, which significantly decrease the stiffness of the concrete element propagate even under insignificant load. The displacements in actual concrete structure as significantly larger than obtained from numerical simulations.

4. Table 9 - it is very difficult to read. It would be useful to underline the most important (interesting) results.

5. I am not sure that the expression "crack displacement" is correct.

6. Conclusions, lines 473-474: "...optimal waterproofing system can be selected." Please, remove the word "optimal". This study did not concern the optimization.

Author Response

The authors would like to express their sincerest gratitude for taking time out of the reviewer’s busy schedule to read the article and provide their valued advice, revision points and comments. As there were numerous points of requested revision by different reviewers, the article has gone through some extensive change since the last version. In the below, the authors have provided as detailed as possible response to the respective comments.

Reviewer 4

Authors design and verified the novel procedure of evaluation the waterproofing systems. The paper is written on a reasonable manner, the figures are clear, but the Reviewer has some remarks, which are presented below:

Comment 1

Introduction, lines 48-50: "Numerous other researches indicate that waterproofing of highway bridges and railroad bridge is a topic of high interest in China, where high performance waterproofing membranes types are regularly tested and reported in the recent research papers" - please add references.

Response 1

Relevant references have been added (please refer to the revised version of the paper lines 50-61, and in the references section, please refer to [3] to [9]).

Comment 2

 The Introduction is very brief. Despite that the Authors clearly define the purpose of the research, in reviewers' opinion the literature review is very poor. Please add the literature about cracks or debondings in concrete structures or bridges.

Response 2

More literature review has been included, and literatures about crack and debonding mechanism has been included as well. Please refer to the revised version of Section 1 (Introduction lines 62 to 80, and in the references section, please refer to [10] to [12]).

Comment 3

Did Authors considered the concrete as homogeneous and linearly elastic? Concrete is brittle material. The initial micro-cracks, which significantly decrease the stiffness of the concrete element propagate even under insignificant load. The displacements in actual concrete structure as significantly larger than obtained from numerical simulations.

Response 3

1) As is mentioned in by the reviewer, the fact that the initial micro-cracks cause higher propagation of the concrete member is a fact (this section has been rewritten). The problem with dynamic stability analysis of railway tracks in Korea is that track bed does not account for cracks during design and model simulation (FEA analysis for crack propagation exist, but institutions such as KORAIL or Korean Railway Network Authority). While this does not significantly affect the results for the track dynamic safety evaluation as constant maintenance on the rails and tracks are conducted, the fact that they do not take into account the significant degree of expected displacement in joint and cracks over time during the design stages result in engineers undermining the importance of high performance waterproofing membranes for sustainability purposes (this is usually the case in China and Korea).

2) Nevertheless, the modelling was conducted in compliance to the regular analysis methods according to the KRC 14030 code, but it must be noted (and is outlined in the paper), that the modelling and the methodology applied in the paper to derive the minimum displacement range for the testing is purely a demonstration. In reality, and if the evaluation system were to be put into actual practice, the displacement range would need to take into account complex forms of degradations (including wheel load induced bending moment, thermal expansion and contraction of concrete, settlement, etc.). However, as this is a proposal to introduce a new evaluation method, the theoretical application and the displacement range derivation is conducted based on domestic (Korean) circumstances. For details on how the revision was made, please refer to section 2.2.1 and 2.2.2, lines 236-302

Comment 4

Table 9 - it is very difficult to read. It would be useful to underline the most important (interesting) results.

Response 4

Table 9 (of the previous version of the paper) has been removed, as it is simply one method of presenting the demonstration evaluation results.

Comment 5

I am not sure that the expression "crack displacement" is correct.

Response 5

The authors are also unsure of the exact technical term to explain the “movement of crack.” But there are several references online that coin this phenomenon with the expression “crack displacement.” (or, as the focus of the testing is centered around this topic, joint displacement is also okay). Just for example, we include this source https://pdfs.semanticscholar.org/fefa/d5551c4b90a5ab4b1143d2278eddf27b7aa2.pdf, where in the second slide, we are discussing about the phenomenon of mode I (opening type) of crack displacement.  Please note that we are not discussing crack propagation in this paper (although the expression can apply just as well) in case this is the term the reviewer is thinking of instead of “crack displacement”

Comment 6

Conclusions, lines 473-474: "...optimal waterproofing system can be selected." Please, remove the word "optimal". This study did not concern the optimization.

Response 6

The word optimal has been removed in the revised Conclusion section. Please refer to Lines 474-476 for details.

Lastly, the paper has been significantly revised since the last version due to the careful review and comments from the other reviewers, and the authors hope that the above points of revision based on the reviewers’ comments are sufficient for reconsideration, and your generous re-review of the paper. Thank you kindly again for taking the time out of your busy schedule to review this paper.

Reviewer 5 Report

The experimental program presented in this paper is interesting, while the theoretical background is frequently questionable. Therefore, the quality of Sections 1 and 2 is significantly lower than that of the following sections. It may be convenient to shorten these introductory parts and put more emphasis on Sections 3 and 4.

Here are some other suggestions:

Line 48

“Numerous other researches indicate that …”. List some of these numerous researches in the bibliography. As a general note, the “References” section is very short for a research article. Properly enrich the “References” section and cite the additional documents where appropriate.

Line 101

“in the following Figure 2 below”. Replace with “in Figure 2”.

Line 111

“A specified list of waterproofing material types is does not exist”. Delete “is”.

Line 156

“the expected concrete expansion can reach up to approximately 1.176× 10–2 m (using equation (1))”. What is this? Is it the linear expansion coefficient? If so, the unit of measurement is not “m”. Also check the unit of measurement in column 5 of table 3.

Table 4

Replace “Bridge sength” with “Bridge length”.

Replace “Gauge sength” with “Gauge length”.

Figure 3

  • Specify which of the two tracks you modeled the dynamic effects for.
  • Zoom in on the legend to make the values readable.
  • Usually, cool colors are associated with compression and warm colors are associated with tension. Since you did the opposite, your color map generates some confusion in these images. It is strongly recommended to modify the color map according to the usual convention.

Line 216

“Next, the deformation results are isolated for the waterproofing layer (Figure 3, c)) and the concrete bridge deck (Figure 3, d)), where upon the stress measured on the waterproofing layer was very minimal (0.22 Mpa) and the concrete deck was minimal as well (0.40 Mpa), and the waterproofing layer should be designed to respond to the stress deformation of the concrete bridge deck as it is an adhered surface to the concrete surface.”. This sentence is not clear, rewrite it.

Line 220

“Based on the analysis results, maximum displacement of approximately 5.415× 10–2 m (using equation 2 and 3 on the modelling data) concrete deck displacement is expected”. It is not clear what this displacement is. Is it a vertical or a horizontal displacement? Where is it calculated?

Figure 4

The assumption on the sign of the forces (tensile and compressive forces) is too simplistic, as it does not take into account the laminate behavior of the three-layer system. The three layers, in fact, do not bend independently of each other, but interact at the interfaces. This changes the distribution of tensile and compressive stresses in the layers relative to the distribution of stresses in the simple bent beam. Therefore, the schematic representation of stresses in Figure 4 is of no use for the study of the degradation conditions. Things are much more complex than illustrated in this figure. Actually, the very understanding of the stress transfer mechanism is questionable: the shear strain originates at the interfaces between the waterproofing layer and the upper/lower layers due to a compatibility condition on the horizontal displacements.

Line 381

“Refer to Figure 5, d”. Is it Figure 6d?

Line 397

“Figure 8 portrays that the performance of the 397 waterproofing membranes decreases marginally as the displacement width increases”. This statement is true for CAS and, to a lesser extent, SAS, while it does not apply to CSC and PUC.

Figure 10

Replace the vertical axis label with “Number of cycles resisted”.

Figure 11

Replace the vertical axis label with “Number of cycles resisted”.

Author Response

The authors would like to express their sincerest gratitude for taking time out of the reviewer’s busy schedule to read the article and provide their valued advice, revision points and comments. As there were numerous points of requested revision by different reviewers, the article has gone through some extensive change since the last version. In the below, the authors have provided as detailed as possible response to the respective comments;

Reviewer 5

The experimental program presented in this paper is interesting, while the theoretical background is frequently questionable. Therefore, the quality of Sections 1 and 2 is significantly lower than that of the following sections. It may be convenient to shorten these introductory parts and put more emphasis on Sections 3 and 4.

Here are some other suggestions:

Comment 1

Line 48 : “Numerous other researches indicate that …”. List some of these numerous researches in the bibliography. As a general note, the “References” section is very short for a research article. Properly enrich the “References” section and cite the additional documents where appropriate.

Response 1

Relevant references have been added (please refer to the revised version of the paper lines 50-61, and in the references section, please refer to [3] to [9]). More literature review about crack and debonding mechanism has been included as well. Please refer to the revised version of Section 1 (Introduction lines 62 to 80, and in the references section, please refer to [10] to [12]).

Comment 2

Line 101 : “in the following Figure 2 below”. Replace with “in Figure 2”.

Response 2

Has been revised accordingly. Please refer to Line 133 of the revised version of the paper.

Comment 3

Line 111: “A specified list of waterproofing material types is does not exist”. Delete “is”.

Response 3

Has been revised accordingly. Please refer to Line 143 of the revised version of the paper.

Comment 4

Line 156: “the expected concrete expansion can reach up to approximately 1.176× 10–2 m (using equation (1))”. What is this? Is it the linear expansion coefficient? If so, the unit of measurement is not “m”. Also check the unit of measurement in column 5 of table 3.

Response 4

The section involving thermal stress load condition has been removed. While this is important to consider thermal expansion and contraction of the concrete bridge deck and incorporate it into deriving the minimum displacement range, too many variables in this study (where the main purpose is to discuss the necessity for a practical evaluation for joint/crack displacement resistance performance of waterproofing membranes for concrete bridge deck and demonstration) is a cause for confusion for the readers. To simplify and solidify the theoretical process, only the discussion of the mechanical degradation due to wheel load has been included in the paper.

Comment 4

Table 4: Replace “Bridge sength” with “Bridge length”.  Replace “Gauge sength” with “Gauge length”.

Response 4

The words were spelling errors. “Sength” has been revised to “Length” (Table 4 has been revised in Section 2.2.1)

Comment 4

Figure 3:  Specify which of the two tracks you modeled the dynamic effects for.  Zoom in on the legend to make the values readable. Usually, cool colors are associated with compression and warm colors are associated with tension. Since you did the opposite, your color map generates some confusion in these images. It is strongly recommended to modify the color map according to the usual convention.

Response 5

More details on the FEM modelling has been included (the modelling parameters have been revised as well). The design parameters, time history function for the dynamic loading conditions, the train (KTX) wheel load specification has been included. It was originally supposed to be the modelling of both tracks operating simultaneously, but since this may cause confusion, only one track has been modelled with the dynamic loading (please refer to the revised version of the paper for details, section 2.2.1). Please refer to the revised sections 2.2.1 to 2.2.2 for details.  With regards to the coloring involved, as the type of stress applied on the model track is compression all throughout, but the warm colors are associated with high values and cool colors are associated with low values. This is the general setting associated with deformation analysis with the MIDAS program. Lines 279-280 has been added to specify this feature of the program.

Comment 6

Line 216: “Next, the deformation results are isolated for the waterproofing layer (Figure 3, c)) and the concrete bridge deck (Figure 3, d)), where upon the stress measured on the waterproofing layer was very minimal (0.22 Mpa) and the concrete deck was minimal as well (0.40 Mpa), and the waterproofing layer should be designed to respond to the stress deformation of the concrete bridge deck as it is an adhered surface to the concrete surface.”. This sentence is not clear, rewrite it.

Response 6

The section has been revised completely (please refer to Response 4 for the reasons). Also please refer to the revised version of the paper for details in Lines 284 to 302 for the rewritten logic on the derivation of minimum displacement range for the demonstration testing of joint displacement resistance performance.

Comment 7

Line 220 : “Based on the analysis results, maximum displacement of approximately 5.415× 10–2 m (using equation 2 and 3 on the modelling data) concrete deck displacement is expected”. It is not clear what this displacement is. Is it a vertical or a horizontal displacement? Where is it calculated?

Response 7

Again, this section has been revised completely for the same reason as the for the comment above (please refer to Response 4 for the reasons). Also please refer to the revised version of the paper for details in Lines 284 to 302 for the rewritten logic on the derivation of minimum displacement range for the demonstration testing of joint displacement resistance performance.

Comment 8

 Figure 4: The assumption on the sign of the forces (tensile and compressive forces) is too simplistic, as it does not take into account the laminate behavior of the three-layer system. The three layers, in fact, do not bend independently of each other, but interact at the interfaces. This changes the distribution of tensile and compressive stresses in the layers relative to the distribution of stresses in the simple bent beam. Therefore, the schematic representation of stresses in Figure 4 is of no use for the study of the degradation conditions. Things are much more complex than illustrated in this figure. Actually, the very understanding of the stress transfer mechanism is questionable: the shear strain originates at the interfaces between the waterproofing layer and the upper/lower layers due to a compatibility condition on the horizontal displacements.

Response 8

This was not intended to describe that the layers in the bridge move bend independently of each other, and as the reviewer aptly describes, it the distribution of the stress at the different interfaces within the layers that must be taken in to consideration (Rodrigues, J., Dias, A., Providencia, P., Timber-Concrete Composite Bridges: State-of-the-Art Review, Bioresources, 2013, 8(4):6630-6649, DOI: 10.15376/biores.8.4.6630-6649

  • Upon inspection, it was determined that the theoretical application of this section does not apply to the focus of the study (as the reviewer mentions), and the Figure 4 (of the older version of the paper) has been removed from the paper. The focus on the FEA modelling methodology has been made stronger, and the rational for the requirement for this evaluation method is established in Section 2.1.
  • Section 2.1) is the theoretical part that describes the necessity of the joint displacement. The section explains the concept on the effect of zero-span tensile stress on the adhered waterproofing membrane on the concrete deck surface in the situation where a crack or joint is present. As is commonly known by the concept of stress concentration factor, when stress is applied to a concrete member, the stress is concentrated at the tip of the crack. When the crack displaces (due to the bending moment of the bridge caused by the wheel load, thermal expansion contraction, or other environmental causes), the localized stress will cause the adhered waterproofing membrane to receive higher stress as well (source: Kunieda, M. Rokugo, K. Recent Progress of HPFRCC in Japan—Required Performance and Applications. J. Adv. Concr. Technol. 2006, 4, 19–33). The common expression to describe the waterproofing membrane’s performance to resist against this stress is called “crack bridging” property (ASTM C 1305 Standard Test Method for Crack Bridging Ability of Liquid-Applied Waterproofing Membrane).
  • It is important to highlight this section as inability to have crack bridging property for waterproofing membranes in railway bridge structures where cracked sections displace constantly will break or allow the formation of leakage path due to cohesive (break within the waterproofing membrane layer) or adhesive (peel off from concrete surface) failure.

Comment 9  

Line 381: “Refer to Figure 5, d”. Is it Figure 6d?

Response 9

It was supposed to be reference to Figure 7 (of the previous version of the paper). This line has been revised accordingly.  (please refer to the revised version of the paper, Line 390)

Comment 10

 Line 397: “Figure 8 portrays that the performance of the 397 waterproofing membranes decreases marginally as the displacement width increases”. This statement is true for CAS and, to a lesser extent, SAS, while it does not apply to CSC and PUC.

Response 10

Result statement regarding CAS has been revised to reflect the reality of the results better. Please refer to the revised paper lines 412 to 420. Also, Figure 8 is now Figure 11 in revised version of the paper.

Comment 11

 Figure 10:  Replace the vertical axis label with “Number of cycles resisted”.

Response 11

Vertical axis label revised accordingly. (Now Figure 12 in the revised paper, please refer to the revised version)

Comment 12:

 Figure 11: Replace the vertical axis label with “Number of cycles resisted”.

Response 12

Figure 11 (of the previous version of the paper) has been removed for clarity of presentation.

Also, it should be noted that the sections, particularly the parts about the 2 separate degradation conditions (thermal stress induced expansion and contraction and wheel load induced bending moment), as well as the equations and calculations, have been consolidated to clarify the presentation of the process of using the proposed evaluation method in real situations. The authors agreed that theories included in the previous version of the paper were too many when all that was need was to clarify just one factor that causes the displacement range to increase. Therefore, more focus and explanation on the methodology to derive the displacement range through a FEA modelling on the bending moment was provided.

Lastly, the paper has been significantly revised since the last version due to the careful review and comments from the other reviewers, and the authors hope that the above points of revision based on the reviewers’ comments are sufficient for reconsideration, and your generous re-review of the paper. Thank you kindly again for taking the time out of your busy schedule to review this paper.

Round 2

Reviewer 1 Report

Comments:

Polymer matrix is a potential waterproof material that can be used in railway bridge deck. The high flexibility of polymer can also minimise crack propagation. Suggest to include this information in introduction section to strengthen the literature review. The following papers can be referred

  1. Optimal design for epoxy polymer concrete based on mechanical properties and durability aspects
  2. Effect of elevated in-service temperature on the mechanical properties and microstructure of particulate-filled epoxy polymers 

Author Response

Dear Reviewer 1

The authors of the article Materials-903292 would like to once again express their sincerest gratitude to the respective reviewers for taking time out of their busy schedule to once again participate in the revision process. We hope we were able to address most of the required amendments to the article, and we would greatly appreciate it if the respective reviewers could take a moment to read over the second revised draft of the manuscript.

Reviewer 1

Comment 1

Polymer matrix is a potential waterproof material that can be used in railway bridge deck. The high flexibility of polymer can also minimise crack propagation. Suggest to include this information in introduction section to strengthen the literature review. The following papers can be referred

  1. Optimal design for epoxy polymer concrete based on mechanical properties and durability aspects
  2. Effect of elevated in-service temperature on the mechanical properties and microstructure of particulate-filled epoxy polymers 

Response 1

The above 2 references have been added and the introduction section (Lines 175 to 178), as reference numbers [22] and [23] respectively. Please refer to the revised paper for details

Thank you kindly for the comments on the revised version of the manuscript. The authors sincerely hope that the second revised version is an improvement to the previous one. Due to your valuable contribution, the article was able to undergo a significant improvement since the first draft. We earnestly appreciate your concern and care for our manuscript.

Reviewer 2 Report

The research area and results are from the context of the manuscript can better understand.
Thank you, for your replies and for editing the manuscript.

Table 1 is on two pages.
Table 4 is on two pages.
Table 6 is on two pages.
Figure 11 is on two pages.

In the solved area is in extensive research.
The part of the Introduction to must be rewritten.
The number of suitable citations is not sufficient.

The part of the Conclusion is too long. Must be rewritten.
It is also useful to list the benefits of the article for further research.
The manuscript must clearly present benefits for further research.

However, in summary, it can be still manuscript significantly improved for the reader.

The document must undergo a minor revision.

Author Response

Dear Reviewer 2

The authors of the article Materials-903292 would like to once again express their sincerest gratitude to the respective reviewers for taking time out of their busy schedule to once again participate in the revision process. We hope we were able to address most of the required amendments to the article, and we would greatly appreciate it if the respective reviewers could take a moment to read over the second revised draft of the manuscript.

Reviewer 2

The research area and results are from the context of the manuscript can better understand.
Thank you, for your replies and for editing the manuscript.

Comment 1

Table 1 is on two pages.
Table 4 is on two pages.
Table 6 is on two pages.
Figure 11 is on two pages.

Response 1

The pagination for the above tables and figure have been revised such that they are no longer in 2 pages. Please refer to the revised paper for details. But please keep in mind that the MDPI allows tables and figures to be in two separate pages. We prepared a draft of both versions and specifically attached a version with page separation (Tables and Figures being in one page), and the line references to the revised sections are in accord with this version.

Comment 2

In the solved area is in extensive research. 
The part of the Introduction to must be rewritten. 
The number of suitable citations is not sufficient.

Response 2

Sections in the introduction section has been revised (lines 35, 41 to 51, and 175 to 178) to include more  suitable citations and clarify the objective of the study. Please refer to the references added below;

  1. Korean National Railway, Railroad Bridge Surface Waterproofing Optimization Plan Research Report, Railway Information Center, 2007, pp 1-148, http://210.90.197.12/jsp/industry/ret/researchTechInfoDetail.jsp?menuId=M010210&objectId=0900271a800ebb66&docId=&p_id2=1107100900271a800ebb66&q_levCd=&q_objectseq=325&q_totalPage=56&q_frPage=24&q_name=1
  2. Bilotti, P.E., Long term deck waterproofing of highway and rail bridges, 2000, https://www.semanticscholar.org/paper/LONG-TERM-DECK-WATERPROOFING-OF-HIGHWAY-AND-RAIL-Bilotti/ae236d97e3aa225bf3d1c4a20061bcff18015599?p2df.
  3. Nielson, D., Chattopadhyay, G., Raman, D., Life Cycle Cost Estimation for Railway Bridge Maintenance, International Heavy Haul Association Conference, 2013, https://www.researchgate.net/publication/273451656_Life_Cycle_Cost_Estimation_for_Railway_Bridge_Maintenance
  4. Kohtbehsara, M.M., Manalo, A., Aravinthan, T., Reddy, K.R., Effect of elevated in-service temperature on the mechanical properties and microstructure of particulate-filled epoxy polymers, Polymer Degradation and Stability, 2019, 10.1016/j.polymdegradstab.2019.108994
  5. Ferdous, W., Manalo, A., Wong, H.S., Abousnina, R., AlAjarmeh, O.S., Yan, Z.G., Schubel, P., Optimal design for epoxy polymer concrete based on mechanical properties and durability aspects, Construction and Building Materials, 2020, https://doi.org/10.1016/j.conbuildmat.2019.117229
  6. Al-Zahrani, M.M., Al-Dulaijan S.U., Ibrahim M., Saricimen, H., Sharif F.M.,Effect of waterproofing coatings on steel reinforcement corrosion and physical properties of concrete

Comment 3

The part of the Conclusion is too long. Must be rewritten. 
It is also useful to list the benefits of the article for further research.
The manuscript must clearly present benefits for further research.

Response 3

The conclusion section has been rewritten to be shorter, and in section 3) of the conclusion (lines 551 to 561), the benefits of the article for further research has been clarified. Please refer to the revised paper for details.

Comment 4

However, in summary, it can be still manuscript significantly improved for the reader.

The document must undergo a minor revision

Response 4

Thank you kindly for the comments on the revised version of the manuscript. The authors sincerely hope that the second revised version is an improvement to the previous one. Due to your valuable contribution, the article was able to undergo a significant improvement since the first draft. We earnestly appreciate your concern and care for our manuscript.

Reviewer 4 Report

Dear Authors,

thank you for your response to my remarks.

Author Response

Thank you kindly for the comments on the revised version of the manuscript. The authors sincerely hope that the second revised version is an improvement to the previous one. Due to your valuable contribution, the article was able to undergo a significant improvement since the first draft. We earnestly appreciate your concern and care for our manuscript.

Reviewer 5 Report

The revised version has been considerably improved, compared to the original one. Many of the causes of confusion for readers have been removed and the focus of the paper is better highlighted. However, some minor and major corrections are still needed (see below). Additionally, this reviewer recommends an extensive review of the English language and sentence structure. Some of the major corrections are listed below, but many more are needed.

Line 13

“thermal stress deformation and stress intensity factor localized at the joints or cracks in the concrete deck are analyzed to derive an expected displacement value”. Since the section involving the thermal stress load condition has been removed, delete any reference to the thermal stress from the abstract section. Additionally, your FEM analysis is performed in the elastic field for a non-cracked bridge deck, which means that there are no stress concentrations at the crack tips. Consequently, the expected displacement value is not derived based on the stress intensity factor localized at the joints. Finally, this sentence is too long. Divide it into two or more sentences to help understanding.

Line 25

Delete any reference to the thermal stress from the keywords.

Line 87

“joint or crack displacement”. Do you mean “joint or crack opening”? Displacement is not a definable parameter for a crack.

Line 126

“the joint/crack resistance performance property of the waterproofing materials”. Rearrange the word order to improve comprehension.

Line 168

“It is proposed that should a proper evaluation system that can adequately evaluate the waterproofing material performance that cater specifically to the degradation conditions, the sustainability of PSC railroad bridges can improve as well.”. Unclear sentence. Reformulate it.

Line 192

“Table 3 below”. Table 3 is actually above the text, not below. As a general recommendation, it would be better to avoid using “below” and “above” with reference to figures and tables (the final pagination may differ from your pagination). Just indicate the numbers of figures and tables.

Line 198

“is concentrated than the surrounding region”. Maybe you mean “is more concentrated than the surrounding region”.

Line 207

“concrete crack displacement”. As above, displacement is not a definable parameter for a crack. Do you mean “concrete crack opening”?

Figure 4 (now deleted)

Since this figure has been deleted, the original comment is irrelevant. However, this reviewer would like to point out that, although the authors’ intention was different, this figure actually depicts the layers as bending independently of each other. In fact, only in the assumption of independent layers the stresses change sign in each layer. The stress sign does not alternate at the interfaces of a laminate.

Line 279

Replace “Figures 4 shows” with “Figure 8 a) shows”.

“lateral displacement perpendicular to the cross section of the bridge structure, the X direction”. Why do you give displacement in the x direction the name of lateral displacement? This is an axial displacement, where the axial displacement is the displacement in the direction of motion.

Line 309

“In order to conduct a joint or crack displacement evaluation method”. Do you mean “crack opening evaluation method”?

Line 317

“The substrate parts (mortar) of Figure 9, a) are mixed at water to cement to sand ratio of 0.4:1:3”. Is this the water to cement to sand ratio recommended by the specifications for railroad bridge decks? The cement to sand ratio of 1:3 does not provide a high quality concrete.

What about the size, shape (rounded or not), origin (river or quarry), and composition of the aggregates? Do they comply with the specifications for the railway bridge deck? If the cement to sand ratio and aggregates do to conform to the specifications, the results of the experimental program may not be representative of the actual behavior of the joints in railroad bridge decks.

Line 321

“Plastic vinyl sheets should be used to cover the molds while curing to prevent evaporation.”. What is the meaning of this sentence? Did you actually use plastic vinyl sheets, or not?

Figure 9

“Crack displacement testing specimen (mortar substrate) structure layout;”. Crack opening testing specimen?

Line 370

crack or joint opening?

Line 373

“lateral displacement”. Replace with axial displacement.

Table 7

Delete any reference to the thermal stress, as you removed this part.

Replace “crack displacement” and “joint displacement” with “crack opening” and “joint opening”, respectively.

Line 416

“Based on the displacement resistance performance results of the 5 specimens for each 416 waterproofing system types in Figure 11”. This is a fragment, not a sentence.

Line 444

“The demonstration of the evaluation method shows it is shown that there is a general trend of decreasing performance”. Revise this sentence.

Line 472

“This study demonstrated a typical FEA modelling of a railway bridge track”. A modeling based on a commercial program cannot be demonstrated. Choose a more appropriate verb or rephrase the sentence.

Line 493

“However, in the given conventional situation of international waterproofing performance evaluation methods, a testing method that simulates the joint displacement while being able to evaluate the watertightness of the installed waterproofing membranes is nevertheless required, as this demonstration shows that even in a non-extreme conditioned testing (without shear stress), the displacement resistance performance difference can be clearly outlined between waterproofing system types.”. This sentence is too long. Divide it into two sentences to help understanding.

Line 500

“There are plans to improve on the application of this evaluation method will have to include more accurate variables that comply to the environmental conditions of the railroad bridge decks along with results of more types of waterproofing systems.”. Review the structure of this sentence.

Author Response

Dear Reviewer 3

The authors of the article Materials-903292 would like to once again express their sincerest gratitude to the respective reviewers for taking time out of their busy schedule to once again participate in the revision process. We hope we were able to address most of the required amendments to the article, and we would greatly appreciate it if the respective reviewers could take a moment to read over the second revised draft of the manuscript.

 Reviewer 3

The revised version has been considerably improved, compared to the original one. Many of the causes of confusion for readers have been removed and the focus of the paper is better highlighted. However, some minor and major corrections are still needed (see below). Additionally, this reviewer recommends an extensive review of the English language and sentence structure. Some of the major corrections are listed below, but many more are needed.

Comment 1

Line 13

“thermal stress deformation and stress intensity factor localized at the joints or cracks in the concrete deck are analyzed to derive an expected displacement value”. Since the section involving the thermal stress load condition has been removed, delete any reference to the thermal stress from the abstract section. Additionally, your FEM analysis is performed in the elastic field for a non-cracked bridge deck, which means that there are no stress concentrations at the crack tips. Consequently, the expected displacement value is not derived based on the stress intensity factor localized at the joints. Finally, this sentence is too long. Divide it into two or more sentences to help understanding.

Response 1

References to the thermal stress variable for obtaining the initial opening range for the testing condition have been removed throughout the paper. Also, as mentioned by the reviewer, the correlation to stress concentration for the initial displacement range (opening range) has been removed.

Comment 2

 Line 25

Delete any reference to the thermal stress from the keywords.

Response 2

References to the thermal stress variable for obtaining the initial opening range for the testing condition have been removed throughout the paper. 

Comment 3

Line 87

“joint or crack displacement”. Do you mean “joint or crack opening”? Displacement is not a definable parameter for a crack.

Response 3

The word displacement has been revised to “opening” throughout the paper as recommended by the reviewer

Comment 4

Line 126

“the joint/crack resistance performance property of the waterproofing materials”. Rearrange the word order to improve comprehension.

Response 4

Revised (please refer to the revised version’s Lines 125~126 )

Comment 5

Line 168

“It is proposed that should a proper evaluation system that can adequately evaluate the waterproofing material performance that cater specifically to the degradation conditions, the sustainability of PSC railroad bridges can improve as well.”. Unclear sentence. Reformulate it.

Response 5

The section has been revised for clarity (please refer to lines 186~189  in the revised version)

Comment 6

 Line 192

“Table 3 below”. Table 3 is actually above the text, not below. As a general recommendation, it would be better to avoid using “below” and “above” with reference to figures and tables (the final pagination may differ from your pagination). Just indicate the numbers of figures and tables.

Response 6

The “bellow” and “above” references have been removed throughout the paper

Comment 7

 Line 198

“is concentrated than the surrounding region”. Maybe you mean “is more concentrated than the surrounding region”.

Response 7

The reviewer is correct, please refer to the revised version Line 216.

Comment 8

 Line 207

“concrete crack displacement”. As above, displacement is not a definable parameter for a crack. Do you mean “concrete crack opening”?

Response 8

The term displacement seems to be used in US literature regarding cracks, but as there has been confusions regarding the usage of this term, the word “displacement” has been revised to “opening” as recommended. The word displacement has been revised to “opening” throughout the paper as recommended by the reviewer (with exception to section 2.2.2, Lines 299 to 317, as the parameter discussed in the FEA modelling is actually displacement)

Comment 9

 Figure 4 (now deleted)

Since this figure has been deleted, the original comment is irrelevant. However, this reviewer would like to point out that, although the authors’ intention was different, this figure actually depicts the layers as bending independently of each other. In fact, only in the assumption of independent layers the stresses change sign in each layer. The stress sign does not alternate at the interfaces of a laminate.

Response 9

Thank you  kindly for the information, it seems the authors were relying on an outdated source or somehow misinterpreted the original source (as this figure is sometimes used in Korean literature to discuss forms of stress/force in the bridge structure).

Comment 10

 Line 279

Replace “Figures 4 shows” with “Figure 8 a) shows”.

 “lateral displacement perpendicular to the cross section of the bridge structure, the X direction”. Why do you give displacement in the x direction the name of lateral displacement? This is an axial displacement, where the axial displacement is the displacement in the direction of motion.

Response 10

Replaced Figure 4 with Figure 8 a). Please refer to Line 299 in the revised version

Comment 11

 Line 309

“In order to conduct a joint or crack displacement evaluation method”. Do you mean “crack opening evaluation method”?

Response 11

The word displacement has been revised to “opening.” Here, we would like to emphasize joint rather than crack as the specimen simulates a joint section rather than a crack section. Refer to line 328 in the revised paper.

Comment 12

 Line 317

“The substrate parts (mortar) of Figure 9, a) are mixed at water to cement to sand ratio of 0.4:1:3”. Is this the water to cement to sand ratio recommended by the specifications for railroad bridge decks? The cement to sand ratio of 1:3 does not provide a high quality concrete.

What about the size, shape (rounded or not), origin (river or quarry), and composition of the aggregates? Do they comply with the specifications for the railway bridge deck? If the cement to sand ratio and aggregates do to conform to the specifications, the results of the experimental program may not be representative of the actual behavior of the joints in railroad bridge decks.

Response 12

As you mentioned, the composition, size, shape and other variables should comply with the specifications for the railway bridge, and this is now mentioned in the revised paper lines 341 to 344. As this is an experimental demonstration of the evaluation method (just to demonstrate how it is done), the initial intent was to make the process understandable but not too complicated such that if other institutions are willing to adopt this method, they can change the variables based on the specific requirement (as is the case with the FEM modelling regime).

 Comment 13

Line 321

“Plastic vinyl sheets should be used to cover the molds while curing to prevent evaporation.”. What is the meaning of this sentence? Did you actually use plastic vinyl sheets, or not?

Response 13

This was actually a mistake, as in Korea, we normally use vinyl sheets and cover the specimens for curing process. We removed this  line in the revised paper to prevent possible confusions.

 Comment 14

Figure 9

“Crack displacement testing specimen (mortar substrate) structure layout;”. Crack opening testing specimen?

Response 14

The word displacement has been revised to “opening” in figure 9. In this regard, Figure 12 has also been revised.

 Comment 15

Line 370

crack or joint opening?

Response 15

The word displacement has been revised to “opening.”

Comment 16

 Line 373

“lateral displacement”. Replace with axial displacement.

Response 16

Please refer to line 411 in the revised paper, the line has been revised to avoid confusion

Comment 17

 Table 7

Delete any reference to the thermal stress, as you removed this part.

Replace “crack displacement” and “joint displacement” with “crack opening” and “joint opening”, respectively.

Response 17

Table 7 has been revised accordingly.

Comment 18

 Line 416

“Based on the displacement resistance performance results of the 5 specimens for each 416 waterproofing system types in Figure 11”. This is a fragment, not a sentence.

Response 18

This line was removed as it is no longer relevant in the content of the paper

Comment 19

 Line 444

“The demonstration of the evaluation method shows it is shown that there is a general trend of decreasing performance”. Revise this sentence.

Response 19

The line has been revised, please refer to the revised paper lines 514 to 515.

Comment 20

Line 472

“This study demonstrated a typical FEA modelling of a railway bridge track”. A modeling based on a commercial program cannot be demonstrated. Choose a more appropriate verb or rephrase the sentence.

Response 20

The line has been revised (line 539). The modelling is an example based on a very specific bridge structure, but as the parameters will change based on where it is used (based on different national standards), we originally used the word “demonstration,” as the opening value 1.5mm is not to be used as a fixed standard for this evaluation method. The authors revised the line accordingly to remove confusion.

 Comment 21

Line 493

“However, in the given conventional situation of international waterproofing performance evaluation methods, a testing method that simulates the joint displacement while being able to evaluate the watertightness of the installed waterproofing membranes is nevertheless required, as this demonstration shows that even in a non-extreme conditioned testing (without shear stress), the displacement resistance performance difference can be clearly outlined between waterproofing system types.”. This sentence is too long. Divide it into two sentences to help understanding.

Response 21

The section has been revised. Please refer to the revised paper lines 546 to 561 for details.

Comment 22

 Line 500

“There are plans to improve on the application of this evaluation method will have to include more accurate variables that comply to the environmental conditions of the railroad bridge decks along with results of more types of waterproofing systems.”. Review the structure of this sentence.

Response 22

The section has been revised. Please refer to the revised paper lines 546 to 561 for details.

Thank you kindly for the comments on the revised version of the manuscript. The authors sincerely hope that the second revised version is an improvement to the previous one. Due to your valuable contribution, the article was able to undergo a significant improvement since the first draft. We earnestly appreciate your concern and care for our manuscript.

Round 3

Reviewer 5 Report

The reviewer sincerely appreciates the effort made by the authors to improve this paper.

Just two last comments:

Lines 344-348, the authors copied part of the reviewer’s comment into the text. It is probably a reminder that the authors forgot to delete. Please review.

Response 18: you claimed to have removed the sentence but, in reality, you did not.

Author Response

The authors would like to sincerely thank the reviewer for once again taking time to spend the efforts to review and assist in the improvement of the manuscript. The authors would also like to apologize for missing the revisions for the previous comments. The following are the response;

The reviewer sincerely appreciates the effort made by the authors to improve this paper.

Just two last comments:

Comment 1

Lines 344-348, the authors copied part of the reviewer’s comment into the text. It is probably a reminder that the authors forgot to delete. Please review.

Response 1

Lines 345-348

As the reviewer had said, the revision was incomplete. The lines have been revised as the following; 

"The substrate parts (mortar) of Figure 9, a) are mixed at water to cement to sand ratio of 0.4:1:3, during the mortar casting (as a note, the substrate part mixture ratio can comply to any required specifications or national standards based on the requirement of the testing, and the following procedure will be based on a demonstration version referencing the Korean National Standard (KS) specification for mortar specimen mixture ratio)."

Comment 2

Response 18: you claimed to have removed the sentence but, in reality, you did not.

Response 2

The authors express their apologies for missing this line of revision (we initially noted it down and thought we removed the sentence, but had not). We took the time to revise the line as the following (lines 513-514);

"The average opening resistance performance by the number of opening resisted result between the 5 specimens (results from Figure 11) were derived and are shown in Figures 12."

Once again, the authors express their gratitude for all the assistance and contribution by the reviewers. Thank you